# FlowPrune: Accelerating Attention Flow Calculation by Pruning Flow Network

**Shuo Xu**
Southeast University
atmxsp01@gmail.com

**Yu Chen**
Southeast University
yu_chen@seu.edu.cn

**Shuxia Lin**
Southeast University
shuxialin@seu.edu.cn

**Xin Geng**
Southeast University
xgeng@seu.edu.cn

**Xu Yang**[*]
Southeast University
xuyang_palm@seu.edu.cn

## Abstract

The Transformer architecture serves as the foundation of modern AI systems, powering recent advances in Large Language Models (LLMs) and Large Multimodal Models (LMMs). Central to these models, attention mechanisms capture contextual dependencies via token interactions. Beyond inference, attention has been widely adopted for interpretability, offering insights into model behavior. Among interpretability techniques, *attention flow* — which traces global information transfer across layers — provides a more comprehensive perspective than single-layer attention maps. However, computing attention flow is computationally intensive due to the high complexity of max-flow algorithms. To address this challenge, we introduce **FlowPrune**, a novel framework that accelerates attention flow analysis by pruning the attention graph before applying max-flow computations. FlowPrune uses the Max-Flow Min-Cut Theorem and two structural properties of Transformer to identify and eliminate non-critical graph regions. It comprises two components: **Edge Pruning**, which removes insignificant attention edges, and **Layer Compression**, which discards layers with minimal contributions to the flow. We conduct extensive experiments on LLaMA and LLaVA to evaluate the robustness and effectiveness of FlowPrune. Our results show that FlowPrune achieves high agreement with the original attention flow in both absolute and relative error metrics, as well as in identifying influential input tokens. Finally, case studies in both NLP and vision domains demonstrate that FlowPrune produces consistent interpretability outcomes as the original Attention Flow, validating its practical utility. The code for this paper is publicly available.

## 1   Introduction

The Transformer architecture [1] has become the foundational backbone of modern AI systems, driving major advances in Large Language Models (LLMs) [2–4] and Large Multimodal Models (LMMs) [5–7]. Although various architectural modifications have since been proposed [8, 9], attention remains the core computational primitive, enabling contextual information flow via token interactions. Because attention weights reflect how information is propagated across tokens, they are frequently used as proxies for interpretability. Despite ongoing debate about their faithfulness as explanations [10, 11], attention weights remain valuable tools for interpreting, diagnosing, and analyzing Transformer behavior across diverse applications [12, 13].

---

[*]Corresponding Author

39th Conference on Neural Information Processing Systems (NeurIPS 2025).

Compared to single-layer attention maps that capture only local, layer-specific token interactions, attention flow offers a global perspective by tracing how information propagates from input to output across all layers. This makes it a more effective diagnostic tool for understanding model behavior. To quantify attention flow, [14] models the entire network as a Directed Acyclic Graph (DAG), where nodes represent token embeddings and edges denote attention weights. Within this framework, attention flow between input and output tokens is formulated as a maximum flow problem computed by the Highest-Level Preflow Push algorithm [15]. Although this approach captures cross-layer information that single-layer maps overlook, its computational complexity of $O(L^{2.5}N^3)$ — for $L$ layers and $N$ tokens — makes it impractical for modern large-scale models.

To reduce computational complexity, [14] proposes Attention Rollout, which approximates attention flow by recursively multiplying layer-wise attention matrices, reducing complexity to $O(LN^3)$ but introduces significant limitations. Specifically, it neglects global capacity constraints and flow conservation principles inherent to max-flow optimization, potentially distorting information propagation paths [14]. As a result, it risks producing misleading explanations that contradict ground-truth flows — an issue critical to safety-sensitive applications like medical diagnosis or legal reasoning [16, 17].

To address this, we propose **FlowPrune**, a novel method that accelerates attention flow computation without heavily distorting the analyses results. FlowPrune prunes the attention graph before applying max-flow algorithms where the pruning operation is designed based on the Max-Flow Min-Cut Theorem [18], stating that the maximum flow between a source and a sink node equals the total capacity of the minimum cut separating them. This implies that graph regions not contributing to the minimum cut can be safely pruned without affecting the final flow. We leverage two key properties of Transformer-based networks to identify such regions: the attention weights are calculated using the softmax function, and the attention graph is strictly layered.

FlowPrune consists of two components: **Edge Pruning** and **Layer Compression**, both designed to reduce the attention graph while preserving its critical flow structure. First, due to the softmax operation, many attention weights are close to zero [19], contributing little to information propagation. These weights can be safely removed without significantly affecting the overall max-flow value. Second, because the attention graph is strictly layered, we localize minimum cut edges to specific layers, allowing for substantial graph reduction. We use a random sampling heuristic to identify layers with fewer minimum cut edges, which are then removed, and new edges are created between adjacent layers to maintain connectivity. These two procedures significantly compress the attention graph, enabling more efficient max-flow computation.

We validate FlowPrune on two sets of experiments. First, we evaluate the fidelity and efficiency of our graph compression by applying it to two widely used models: the unimodal Llama and the multimodal LLaVA [20, 21]. We compare the max-flow computation on the original and compressed graphs in terms of runtime and absolute/relative error. Since attention flow is often used to assess which input tokens contribute more to model predictions [14], we also measure the ranking discrepancy of input tokens by comparing their relative max-flow orderings. Second, we conduct case studies on real-world scenarios where attention flow has been applied for model analysis. Specifically, we test the practicality of FlowPrune in two real-world scenarios: analyzing Paraphrasing Verification [22] and constructing ViT Heatmaps, where results prove that FlowPrune is more similar with the original Attention Flow than Attention Rollout, validating its practicality. All our code is publicly available, and you can find it at https://github.com/ATMxsp01/FlowPrune.

## 2 Related Work

**Attention Flow and Interpretability.** The interpretability of attention mechanisms has attracted significant interest, particularly in NLP and vision models. While some studies suggest that attention weights offer insight into model reasoning [12, 23, 24], others argue that they may not faithfully reflect decision-making processes [10, 11, 25]. Despite this debate, attention remains a widely used tool for attributing model predictions to input elements. For instance, attention-based visualizations have been used to identify important areas in image classification [26, 27], diagnose linguistic errors in translation [28, 29], and trace causal relationships in sentence classification [10, 30]. Beyond direct visualization, recent work has formalized attention flow as a global explanation mechanism [14], tracing information propagation across layers via graph-based analysis. However, such methods are computationally expensive and may become infeasible for large models. Several approximations,

including Attention Rollout [14], have been proposed to reduce complexity, though they often sacrifice theoretical properties such as flow conservation. Meanwhile, the observed sparsity of attention matrices [19] motivates efficient pruning-based approaches that preserve interpretive fidelity while improving scalability.

**Maximum Flow Algorithms.** The maximum flow problem is a classical topic in graph theory, which seeks to determine the maximum amount of flow that can be sent from a source to a sink under capacity constraints. Common algorithms include Edmonds–Karp [31], which uses breadth-first search to find augmenting paths, Dinic's algorithm [32], which improves throughput via level graphs, and Push–Relabel method [33], which maintain a preflow and iteratively update local excess. Among these, the *Highest-Level Preflow-Push* (HLPP) algorithm [15] has demonstrated strong practical efficiency, especially on layered or DAG-structured graphs, making it suitable for attention flow computation [14].

Building on this foundation, our method, **FlowPrune**, accelerates attention flow computation by applying HLPP to a structurally simplified attention graph. We leverage the Max-Flow Min-Cut Theorem [18] to guide pruning decisions, safely removing low-capacity edges and compressible layers that do not contribute to the minimum cut. This design reduces computational complexity while preserving critical attribution paths, enabling scalable analysis of large Transformer models.

# 3 FlowPrune

To address the scalability limitations of attention flow analysis in large-scale Transformer networks, we propose an efficient approximation method named **FlowPrune**. In this section, we first review the standard Attention Flow formulation and its computational bottlenecks. We then introduce two key approximation strategies, namely **Edge Pruning** and **Layer Compression**, which jointly reduce time complexity while preserving essential attention flow structures to calculate maximum flow.

## 3.1 Calculating Maximum Flow

To calculate the maximum flow that quantifies how attention propagates across layers in a $L$-layer Transformer, [14] models the attention mechanism as a layered directed graph $\mathcal{G} = (\mathcal{V}, \mathcal{E})$. Specifically, $\mathcal{V} = \{\mathcal{V}_1, \ldots, \mathcal{V}_L\}$, where each $\mathcal{V}_i$ represents the set of nodes corresponding to token representations at the $i$-th Transformer layer. The edge set is defined as $\mathcal{E} = \{\mathcal{E}_1, \ldots, \mathcal{E}_{L-1}\}$, where $\mathcal{E}_i$ denotes directed edges capturing attention from tokens in layer $i$ to tokens in layer $i + 1$, with normalized attention weights used as edge capacities.

Given this graph $\mathcal{G}$ constructed from an $L$-layer Transformer, the classical Highest-Level Preflow Push (HLPP) algorithm can be applied to compute the maximum flow, with a time complexity of $O(V^2\sqrt{E})$. Since the number of nodes and edges scale as $V \approx LN$ and $E \approx N^2L$ respectively — where $N$ is the number of tokens per layer and $L$ is the number of layers — the total computational complexity becomes $O(L^{2.5}N^3)$. This high cost makes the method impractical for large-scale Transformer models.

## 3.2 Pruning the Attention Graph based on Max-Flow Min-Cut Theorem

Our objective is to accelerate the computation of maximum flow in large-scale transformer architectures without largely disturbing the analyses results. To achieve this, we brought ideas from the classic **Max-Flow Min-Cut Theorem** [18] theory, which states that: In a directed graph with non-negative edge capacities, the maximum amount of flow that can be sent from a source node to a sink node equals the total capacity of the minimum cut that separates the source from the sink. This theory provides a useful intuition: if certain regions of the graph do not have the minimum cut, they can be safely removed without significantly changing the overall flow, e.g., by pruning low-capacity edges or compressing intermediate layers without minimum cut.

To apply the above-mentioned insight to prune the attention graph, we also need to exploit two key structural properties of Transformer. First, the computation of attention weights involves a softmax operation and second, Transformer exhibits a strictly layered structure. We next show how to integrate these two properties with Max-Flow Min-Cut Theorem to design **Edge Pruning** and **Layer Compression** strategies for reducing the graph scale.

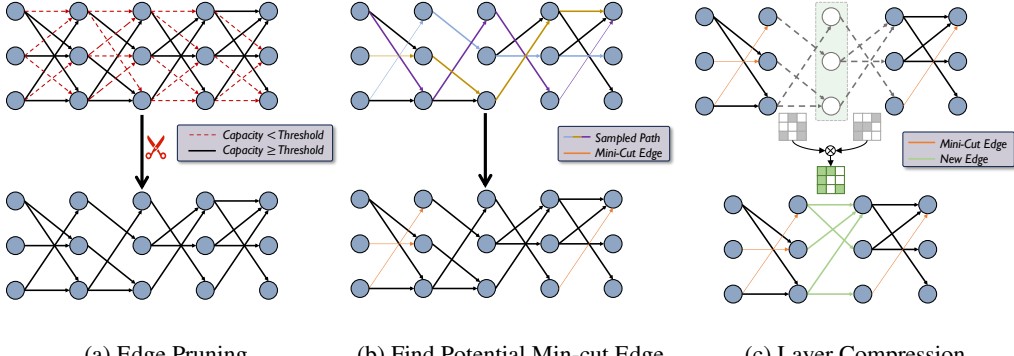

(a) Edge Pruning.     (b) Find Potential Min-cut Edge.     (c) Layer Compression.

Figure 1: Pipeline of FlowPrune. (a) The edges with low flow capacity are removed. (b) Three paths are sampled (blue, purple, and brown ones) to find the layers which has less mini-cut edge, where the third layer does not contain min-cut edges. (c) The third layer is removed and the connectivity between the second and fourth layers are re-built by multiplying two binary adjacency matrices.

**Edge Pruning.** In Transformers, the softmax operation used in attention computation often produces many near-zero attention weights, which contribute minimally to the propagation of information across layers. These negligible weights correspond to edges in the attention graph that carry insignificant flow and can therefore be safely pruned to reduce computational overhead. To eliminate such uninformative edges, as shown in Figure 1a, we set a pruning threshold $\theta$ and remove all edges whose capacities fall below this value, effectively treating them as having zero capacity.

**Layer Compression.** As previously discussed, the attention graph of a Transformer is a strictly layered structure, ensuring all flow paths follow layer-wise transitions. This property offers two key advantages for compression. First, it allows to localize minimum cut edges to specific layers, allowing to treat each layer as a fundamental unit for compression. By removing layers with less critical contributions to the overall attention flow, we can achieve substantial graph reduction while not heavily damaging the maximum flow approximation. Second, the layered structure permits efficient removal of entire intermediate layers and reconstruction of cross-layer connections via matrix multiplication. Next, we show how to identify compressible layers and establish new cross-layer connections.

Motivated by the Max-Flow Min-Cut Theorem, we seek to compress layers containing edges that minimally contribute to the network's minimum cut, as these are less critical for information flow. Since exact minimum cut computation is computationally expensive — equivalent to solving the full max-flow problem [14] — we introduce an efficient sampling-based heuristic, which is shown in Figure 1b. After constructing the attention graph, we randomly sample source-to-sink paths in this graph and identify the lowest-capacity edge in each path as a *potential min-cut edge*. Then we compress layers containing relatively few such edges. Following Transformer architecture principles [13], we preserve the first and last layers to maintain essential signal initiation and termination.

It should be noted that identifying the lowest-capacity edge along a sampled path as a potential min-cut edge is a heuristic rather than a guaranteed criterion. The intuition is as follows: in the case where a source-to-sink path does not overlap with others, the smallest-capacity edge on this path must belong to the minimum cut by definition [18]. However, in more realistic settings where multiple paths often share edges, this assumption becomes less precise. Nevertheless, attention graphs derived from Transformers are densely connected in a strictly layered structure, where each token in layer $i$ typically attends to all tokens in layer $i + 1$. This near-uniform connectivity implies that sampled paths are equally likely to overlap on any edge. As a result, the edge with the lowest capacity on a random path remains statistically more likely to lie on the global min-cut, justifying its use as a proxy in our sampling-based approximation.

After identifying layers for compression, we reconnect the remaining layers to maintain attention flow connectivity. When compressing layer $i$, we add direct edges from nodes in layer $i$ to layer $i + 2$ whenever a path exists through layer $i + 1$. In practice, this reconnection can be efficiently implemented by multiplying binary adjacency matrices representing layer-wise connectivity, as illustrated in Figure 1c. The new edge capacities are set to 1, which preserves the maximum flow

since: (1) all original attention weights are normalized to $[0, 1]$, and (2) flow network theory guarantees that increasing edge capacities beyond actual flow values maintains the maximum flow [34].

Combining edge pruning and layer compression significantly reduces the attention graph size. We then compute the maximum flow using Highest-Level Preflow Push algorithm on this compressed graph, achieving substantial complexity reduction while maintaining flow accuracy.

# 4 Experiments

We assess the effectiveness of FlowPrune based on computational efficiency, fidelity to the original attention flow, and practical utility in real-world tasks. We evaluate FlowPrune in two cases: (1) generation cases on LLMs and LMMs, and (2) application cases on language and vision tasks. All our experiments were conducted on a single server equipped with 2 AMD EPYC 7453 28-Core Processors and 4 NVIDIA RTX A6000 GPUs.

## 4.1 Fundamental Evaluation

### 4.1.1 Experimental Settings and Evaluation Metrics

We evaluate **Llama** [20] and **Qwen** [35], representing language-only transformers, on the other hand **LLaVA** [21] and **QwenVL** [36] representing vision-language transformers, respectively. Each model has 32 layers. We experiment with seven compression configurations: the full model (32 layers), an extreme compression setting (3 layers), and five intermediate retain rates of $80\%, 75\%, 50\%, 30\%$, and $25\%$, which correspond to 26, 24, 16, 10, and 8 layers, respectively. The edge pruning threshold is set to $1 \times 10^{-6}$ throughout all experiments.

The datasets used by these models to generate attention maps are GSM8K [37] and OKVQA [38], respectively. For each model, we provided 1,000 inputs and randomly selected 100 token pairs from each generated attention map (including those compressed using FlowPrune) to calculate the corresponding results.

We propose four fundamental metrics to assess the efficiency and approximation quality. **(1) Computational Cost (Time).** This metric measures the wall-clock time for computing attention flow using FlowPrune, compared to the original implementation, serving as an indicator of computational efficiency. Lower values indicate faster inference. **(2) Approximation Error.** We compute both the absolute and relative errors between the approximated and original attention flow matrices, averaged across all examples. Smaller values reflect higher accuracy in the approximation. **(3) Top-$K$ Token Retention.** For each input, we extract the top-$K$ highest-flow tokens from both the original and approximated attention flows, and compute their Jaccard overlap [39]. Higher values indicate better retention of key tokens. Specifically, $IOU = \frac{|A \cap B|}{|A \cup B|}$, where $A/B$ respectively represents the top-$K$ token pairs of the attention flows calculated by original/compressed graphs, where $K = 1, 3, 5, 10, 15, 20, 25, 50$ in the experiments. **(4) First Rank Divergence.** We identify the first position in the sorted top-$K$ token rankings where the approximation diverges from the original output, normalizing this position by sequence length. Higher values reflect better consistency with the original ranking.

### 4.1.2 Results Analysis

**Computational Cost (Time).** Figure 2 presents the speed-up ratios of Llama and LLaVA under varying retain rates. We use 1,000 samples from the **GSM8K** dataset for Llama and 1,000 samples from **OKVQA** for LLaVA. Each sample corresponds to a unique attention graph, and for each graph, attention flows are computed between 1,000 token pairs. The total computation time is used to derive the speed-up ratios.

As previously discussed, for a Transformer with $L$ layers and $N$ tokens per layer, the computational complexity of attention flow computation is $O(L^{2.5}N^3)$. With **Layer Compression**, retaining only $L/R$ layers reduces the complexity to $O(L^{2.5}N^3/R^{2.5})$, achieving a theoretical speed-up of $R^{2.5}$. The empirical results in Figures 2 and 3 generally align with this analysis. For instance, with a 50% retain rate, Llama's computation time is halved-down from 0.55 seconds per case. When compressed to just three layers, FlowPrune requires only 5.31% of the original attention flow time. On LLaVA,

FlowPrune achieves even greater gains, reducing computation time to approximately 35% of the original at a 50% retain rate.

Interestingly, the two models exhibit different speed-up patterns. LLaVA's performance closely matches the theoretical expectation, whereas Llama demonstrates a more linear reduction in average computation time. This discrepancy may be attributed to differences in the attention weight distributions across the two models. Llama may retain more edges during Layer Compression, which causes its time cost to decrease more slowly than the theoretical estimate.

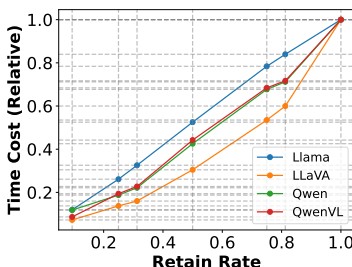

Figure 2: **Speed-up Ratio.** The horizontal axis shows the retain rate (fraction of layers retained), and the vertical axis shows the average time cost normalized by the original Attention Flow runtime, measured over 100 token pairs.

While compressing the attention graph introduces some overhead, this step is performed only once per graph. All subsequent token-pair computations benefit from the reduced graph size. Figure 3 shows the average per-pair computation time — including compression overhead — as the number of token pairs per graph increases. Due to longer input sequences (approximately 500 tokens), LLaVA has a higher average time cost than Llama (typically around 100 tokens). Nonetheless, the compression overhead is amortized as the number of token pairs increases. For a 50% retain rate, the overhead becomes negligible after computing more than 6 cases for Llama and 2 cases for LLaVA.

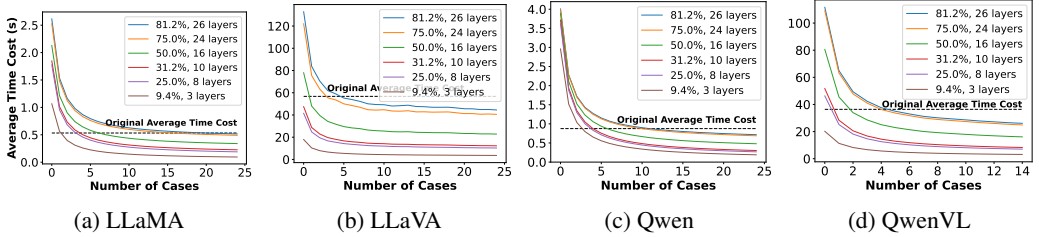

| (a) LLaMA | (b) LLaVA | (c) Qwen | (d) QwenVL |

Figure 3: **Comparison of Average Overhead.** The horizontal axis indicates the number of token pairs computed by per attention map. The vertical axis shows the average time cost per token pair, including the overheads. Each curve corresponds to a specific compression setting. The black dashed line denotes the average computational time of the original (uncompressed) Attention Flow.

**Approximation Error.** Figures 4 provide detailed analyses of how absolute and relative errors vary with the true attention flow values. Since relative error fluctuates with both retain rates and original attention flow values, we present results for all retain rates in Figures 4b and 4e, and separately report results for retain rates of $30\%$ and above in Figures 4c and 4f.

We observe that both absolute and relative errors decrease as the original attention flow approaches 1. This is due to two properties: (1) Since the sum of all edge capacities from a source node in an attention graph is 1, the maximum flow cannot exceed 1. Edge Pruning only reduces edges originating from a node, so this property remains valid for the compressed graph. (2) In Layer Compression, newly added edge capacities are set to 1, which can be regarded as positive infinity in the attention graph, while connectivity between remaining nodes stays unchanged. This ensures that the maximum flow from the compressed graph increases, not decreases.

Both properties constrain the errors when the original attention flow is close to 1, leading to reduced approximation errors. This property is generally favorable, as we are primarily concerned with the top source–destination pairs with the highest attention flow values. These pairs indicate the highest attention weights between token pairs, which are the focus of conventional attention visualization. The small error in this range, as observed with FlowPrune, enhances confidence in its results.

**Top-$K$ Token Retention.** Many studies use attention flow methods focus on the *top-$K$ token pairs with the highest attention flow*. To address this, we use the Top-$K$ Token Retention metric to calculate the Jaccard overlap (IOU) of these token pairs. Figure 5 shows the results. In Figure 5, the horizontal

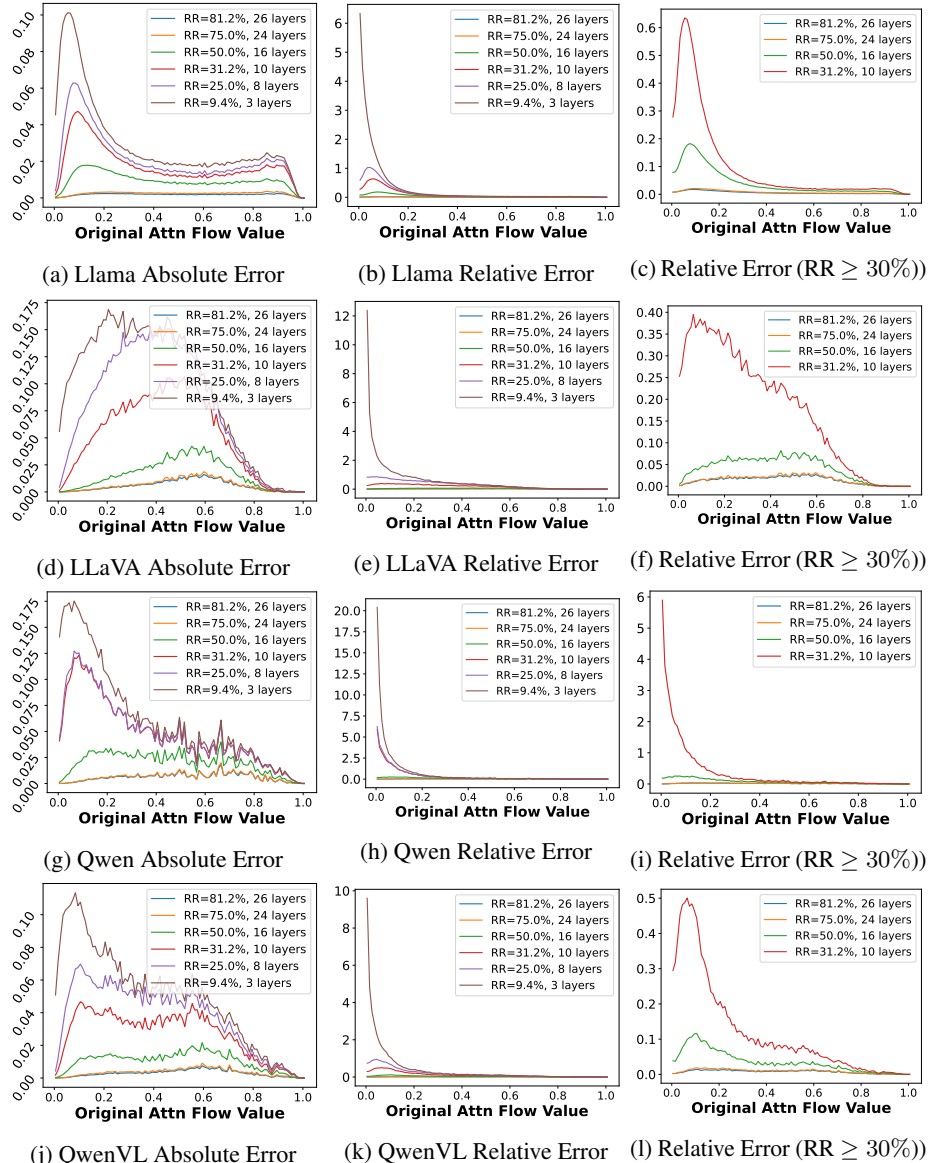

Figure 4: **Approximation Error vs. True Attention Flow Value.** Each subplot shows the approximation error (either absolute or relative) between FlowPrune and the original Attention Flow. The horizontal axis denotes the original attention flow values, and the vertical axis shows the average error magnitude for all token pairs falling within each value. Subplots (a)–(c) show the absolute and relative errors for Llama, and subplots (d)–(f), (g)-(i),(j)-(l) show the same for LLaVA, Qwen and QwenVL. RR represents retain rates.

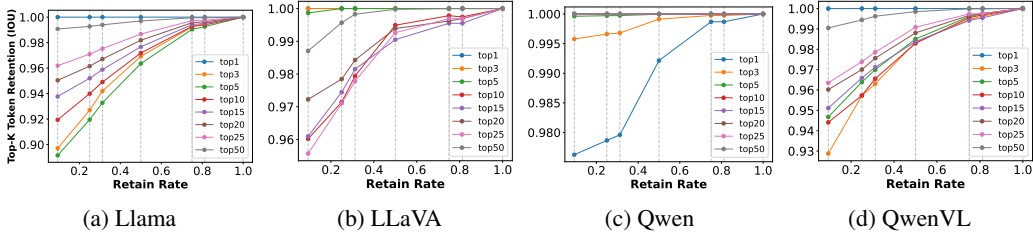

|  |  |  |  |
|:---:|:---:|:---:|:---:|
| (a) Llama | (b) LLaVA | (c) Qwen | (d) QwenVL |

Figure 5: **Top-$K$ Token Retention.** We evaluate the overlap between the top-$K$ tokens selected by FlowPrune and those selected by the original Attention Flow method for different values of $K$. Retention is measured using the Jaccard index (IoU), defined as IoU $= \frac{|A \cap B|}{|A \cup B|}$. Higher values indicate better preservation of salient attention targets.

axis represents the retain rate, defined as the ratio of layers in the compressed attention map to those in the original map. A lower retain rate corresponds to fewer remaining layers. The vertical axis shows Jaccard overlap values. The leftmost point represents the case where the retain rate is minimal, corresponding to compressing the attention map to three layers. Even in this case, the Top-$K$ Token Retention remains around $0.90$, demonstrating that our approximation algorithm maintains high fidelity for the top token pairs. This result aligns with findings from the Approximation Error experiment, which examined error variation with original attention flow values.

**First Rank Divergence.** Figure 6 shows the results of this metric for two models. The curves depict the average First Rank Divergence (FRD) across all attention map samples under varying retain rates. When the retain rate is restricted to 50%, the average FRD of both models exceeds $0.4$. Notably, even at the highest compression level (retaining only 3 layers), the average FRD of Llama is close to $0.2$, while the average FRD of LLaVA exceeds $0.3$. This result is fully sufficient for practical applications with lower precision requirements. While the results at a retain rate 50% are adequate for tasks with higher precision demands.

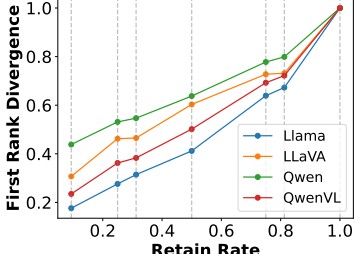

Figure 6: **First Rank Divergence.** This metric records the first position in the sorted top-$K$ token list where the approximation diverges from the original output, normalized by the sequence length.

### 4.2 Application Scenarios

We use two application scenarios in both NLP and Vision, which are **Paraphrasing Verification** and **Vision Transformer (ViT) Heatmaps**, to show the practicality of FlowPrune. Paraphrase Verification [22] involves determining whether two sentences express the same or similar meaning. Attention Flow offers a visual analytics framework for examining self-attention patterns, illustrating how fine-tuning sharpens attention toward task-relevant details. In ViT, attention heatmaps depict the spatial distribution of attention weights, highlighting the model's focus regions during inference.

#### 4.2.1 Paraphrasing Verification

In this task, we need to trace the attention flow from the classification token in the final layer backward through the network, across layers and attention heads. By comparing attention patterns between pre-trained and fine-tuned models, it reveals how fine-tuning sharpens attention on task-relevant distinctions, such as phrases that differentiate sentence meaning. This visual comparison offers valuable insights into the model's reasoning and decision-making process. Here, we replace the original Attention Flow algorithm with FlowPrune and compare its results against those of Attention Flow and Attention Rollout, demonstrating the practical efficiency and effectiveness of FlowPrune.

In this case, we implement the experiments on **MRPC** (Microsoft Research Paraphrase Corpus) [40] dataset with a 12-layer BERT model, which asks the model to determine whether two sentences have the same meaning. We compare the results of FlowPrune and Attention Rollout with the original

Table 1: **Max-$n$ Overlap for Paraphrasing Verification.** FP represents Flowprune and AR represents Attention Rollout.(Mean $\pm$ SE)

| Max-$n$ | FP Pretrained | AR Pretrained | FP Fine-tuned | AR Fine-tuned |
|---------|---------------|---------------|---------------|---------------|
| $n = 3$ | $98.50 \pm 0.86\%$ | $63.40 \pm 2.58\%$ | $95.00 \pm 1.50\%$ | $84.50 \pm 2.32\%$ |
| $n = 5$ | $97.93 \pm 1.19\%$ | $84.71 \pm 1.83\%$ | $92.50 \pm 2.26\%$ | $76.62 \pm 1.93\%$ |
| $n = 10$ | $97.69 \pm 1.33\%$ | $66.38 \pm 1.58\%$ | $91.45 \pm 2.58\%$ | $70.85 \pm 1.56\%$ |

(a) FlowPrune (Pretrained)   (b) Attention Flow (Pretrained)   (c) Attention Rollout (Pretrained)

(d) FlowPrune (Fine-tuned)   (e) Attention Flow (Fine-tuned)   (f) Attention Rollout (Fine-tuned)

Figure 7: **Paraphrasing Verification Example.** Visualization of attention flow results across three methods—FlowPrune, original Attention Flow, and Attention Rollout—on a sentence-pair classification task. The two input sentences are: *(1) "The pound also made progress against the dollar, reached fresh three-year highs at $1.6789."* and *(2) "The British pound flexed its muscle against the dollar, last up 1 percent at $1.6672."*

Attention Flow in both pre-trained and fine-tuned models. In this task, we primarily focus on the attention metrics from tokens in the initial layer pointing to the *[cls]* token in the final layer. The tokens with the highest computed attention metrics are considered the most important.

We calculate the **Max-$n$ Overlap** between the most important tokens obtained using Flow-Prune/Attention Rollout and the original Attention Flow. For FlowPrune, we use a retain rate of $67\%$, reducing the original $12$-layer attention map to $8$ layers. Similar to the Top-$K$ Token Retention, we also use the Jaccard overlap (IOU) as the corresponding metric. We take $n = 3, 5, 10$ and calculate the corresponding results. Table 1 shows the results, where we can see that FlowPrune has higher Max-$n$ Overlap with original Attention Flow than Attention Rollout, proving FlowPrune is a better approximation than Attention Rollout in analyzing the model behaviour. We also show one example from MPRC in Figure 7. We find that the analysis results from FlowPrune are almost identical to those from the original Attention Flow, and both are capable of identifying the focal points of attention, effectively distinguishing between pretrained and fine-tuned models. In contrast, the Attention Rollout method is limited by its inherent flaws, resulting in inferior outcomes compared to the other two analysis methods.

### 4.2.2  Vision Transformer (ViT) Heatmaps

When conducting an attention interpretability analysis for the ViT model, we may be interested in understanding the attention relevance of a specific token to all other tokens to construct an attention heatmap. To test whether FlowPrune is useful here, we use the **Deit-Small** [41] model and implement the experiments on **ILSVRC2012** [42], an image classification dataset. Similarly, we primarily focus on the attention metrics from tokens in the initial layer pointing to the *[cls]* token in the final layer, and compare the relevant analysis results of Attention Flow, FlowPrune, and Attention Rollout. For FlowPrune, we use a retain rate of $25\%$, reducing the $12$-layer attention map to $3$ layers.

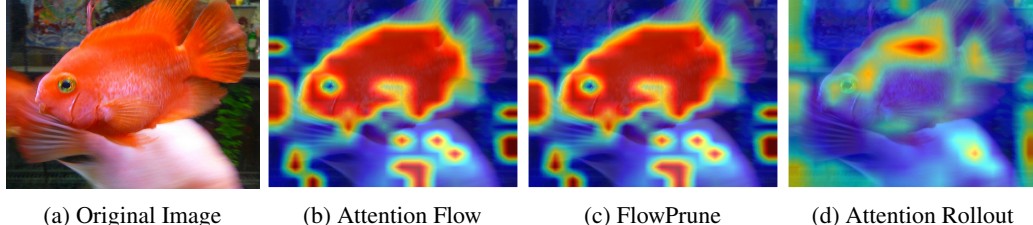

| (a) Original Image | (b) Attention Flow | (c) FlowPrune | (d) Attention Rollout |

Figure 8: **ViT Heapmap Example.** We visualize the attention attribution maps produced by three methods — FlowPrune, the original Attention Flow and Attention Rollout — on a ViT model. All heatmaps correspond to the [CLS] token and show which regions of the image contribute most to the model's final prediction.

Like Paraphrasing Verification, we calculate the **Max-$n$ Overlap** between Original Attention Flow and FlowPrune/Attention Rollout. Considering that ViT has more input tokens, We take $n = 10, 20, 30$ and get results in Table 2. We can see FlowPrune has a much higher Max-$n$ Overlap with the original Attention Flow than Attention Rollout, indicating FlowPrune is a better approximation than Attention Rollout in constructing the heatmaps for ViT. Figure 8 presents an example of a ViT heatmap, which was gener-

Table 2: **Max-$n$ Overlap for ViT Heatmap.** We report the top-$n$ region overlap between FlowPrune (FP), Attention Rollout (AR), and the reference attention map computed via full Attention Flow. (Mean ± SE)

| Max-$n$ | FP | AR |
|---|---|---|
| $n = 10$ | $100.0 \pm 0.00\%$ | $28.18 \pm 3.06\%$ |
| $n = 20$ | $99.53 \pm 0.12\%$ | $50.84 \pm 3.74\%$ |
| $n = 30$ | $99.59 \pm 0.11\%$ | $78.52 \pm 3.43\%$ |

ated using three different methods. **Deit-Small** model utilizes the *[CLS]* token for image classification tasks. Therefore, we focus on the attention metrics from all tokens to the *[CLS]* token to construct heatmaps. The analysis results of FlowPrune once again align with the original Attention Flow, while the Attention Rollout method, due to its inherent flaws, fails to effectively analyze the propagation of attention.

## 5 Conclusion and Limitation

In this work, we propose **FlowPrune**, a novel framework to accelerate attention flow computation, aiming to facilitate the analysis of Transformer-based networks. Motivated by the Max-Flow Min-Cut Theorem, FlowPrune leverages two structural properties of Transformers: the use of the softmax operation in attention computation and the strictly layered architecture of attention graphs. FlowPrune introduces two key components: **Edge Pruning**, which removes insignificant attention edges, and **Layer Compression**, which discards layers with minimal contributions to the flow. Together, these techniques effectively reduce the scale of the attention graph while preserving the fidelity of attention flow analysis. Our method provides a practical and efficient solution for scalable interpretability in large Transformer models, and holds promise for guiding further analysis and application of Large Language Models [43] [44] [45].

While **FlowPrune** effectively accelerates attention flow computation with minimal loss in accuracy for most cases, its relative error tends to correlate with the magnitude of the original attention flow values. In particular, when the original flow value is very small, the relative error introduced by FlowPrune can become excessively large, which may render the results unreliable for interpretability in those regions. This suggests that caution should be exercised when analyzing low-magnitude flows, as the pruning process may disproportionately distort their values.

## 6 Acknowledgment

This work is supported by the National Science Foundation of China (62576091). This research work is also supported by the Big Data Computing Center of Southeast University. This work was also partially supported by the Southeast University Kunpeng & Ascend Center of Cultivation and the Big Data Computing Center of Southeast University.

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

# A Supplementary Materials

To provide additional qualitative evidence for the effectiveness of the proposed FlowPrune framework, this appendix presents five supplementary examples for both **Paraphrasing Verification** and **Vision Transformer (ViT) Heatmaps** Analysis in Section 4. We also included an application scenario experiment on **Question Answer Verification** [22], which is also based on BERT. In addition, we provide a detailed description of the dynamic programming algorithm used to determine which attention layers to compress in FlowPrune.

## A.1 Paraphrasing Verification

To further validate the effectiveness of FlowPrune in interpretability tasks, we conduct additional experiments on the MRPC dataset for paraphrasing verification. This task involves determining whether two input sentences convey the same meaning. For example, the sentence pair *[CLS] It affected earnings per share by a penny. [SEP] The company said this impacted earnings by a penny a share. [SEP]* is labeled as a paraphrase (positive example), since both sentences express semantically equivalent content.

In this setting, we compute attention flow values between token pairs from both sentences using FlowPrune, and compare the results with those obtained from the original Attention Flow and Attention Rollout algorithms. The analysis is performed using both the pre-trained and fine-tuned BERT models. FlowPrune is applied to reduce the computational complexity while preserving essential interpretability signals.

Figures 9 through 13 show five representative examples of the resulting attention heatmaps. As demonstrated, FlowPrune consistently aligns with the original Attention Flow in highlighting key semantic alignments and distinguishing features between the sentences. Notably, the differences between pre-trained and fine-tuned models are also clearly reflected in the FlowPrune heatmaps, showcasing its ability to retain meaningful interpretability insights under compression.

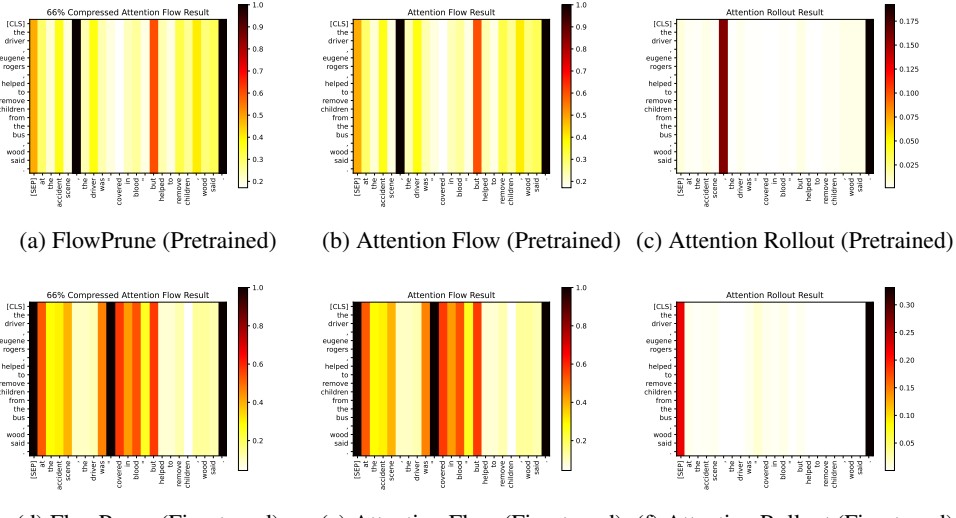

(a) FlowPrune (Pretrained)  (b) Attention Flow (Pretrained)  (c) Attention Rollout (Pretrained)

(d) FlowPrune (Fine-tuned)  (e) Attention Flow (Fine-tuned)  (f) Attention Rollout (Fine-tuned)

Figure 9: **Paraphrasing Verification Example 1.** The two input sentences are: *(1) "The driver , Eugene Rogers , helped to remove children from the bus , Wood said ."* and *(2) "At the accident scene , the driver was " covered in blood " but helped to remove children , Wood said ."*

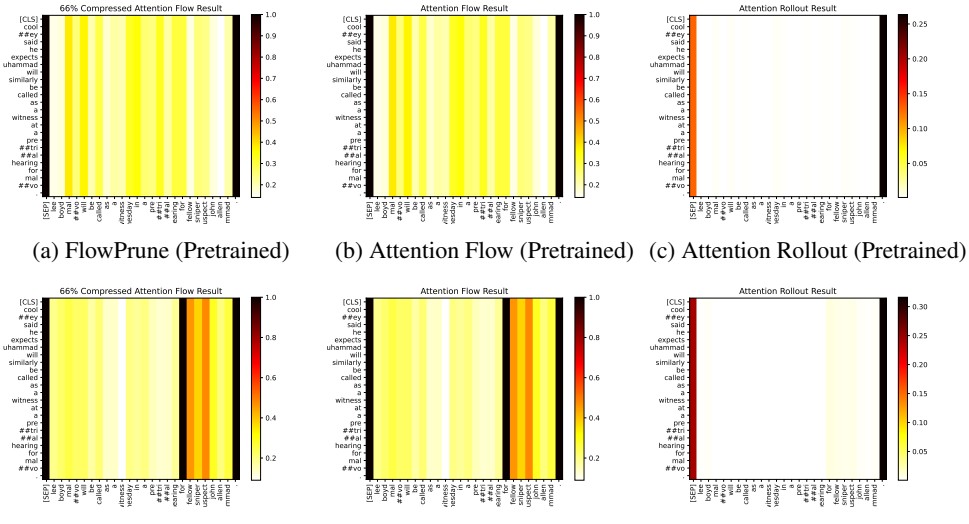

Figure 10: **Paraphrasing Verification Example 2.** The two input sentences are: *(1) "Cooley said he expects Muhammad will similarly be called as a witness at a pretrial hearing for Malvo ."* and *(2) "Lee Boyd Malvo will be called as a witness Wednesday in a pretrial hearing for fellow sniper suspect John Allen Muhammad ."*

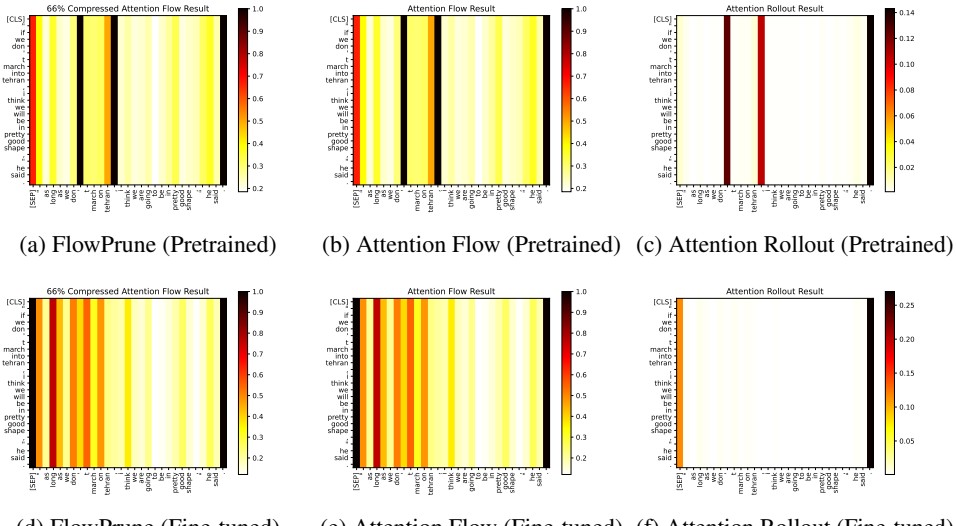

Figure 11: **Paraphrasing Verification Example 3.** The two input sentences are: *(1) "" If we don 't march into Tehran , I think we will be in pretty good shape , " he said ."* and *(2) "" As long as we don 't march on Tehran , I think we are going to be in pretty good shape , " he said ."*

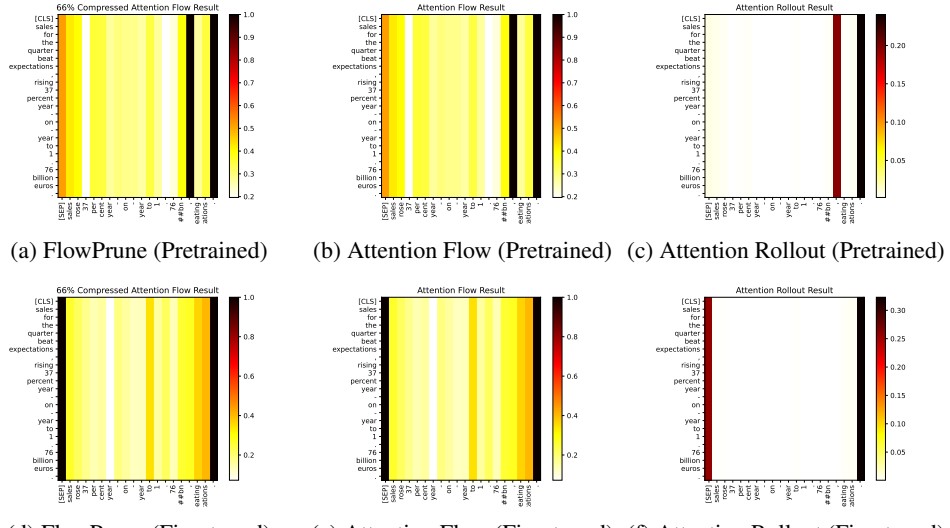

(a) FlowPrune (Pretrained)    (b) Attention Flow (Pretrained)   (c) Attention Rollout (Pretrained)

(d) FlowPrune (Fine-tuned)    (e) Attention Flow (Fine-tuned)   (f) Attention Rollout (Fine-tuned)

Figure 12: **Paraphrasing Verification Example 4.** The two input sentences are: *(1) "Sales for the quarter beat expectations , rising 37 percent year-on-year to 1.76 billion euros ."* and *(2) "Sales rose 37 per cent year-on-year to 1.76bn , beating expectations ."*

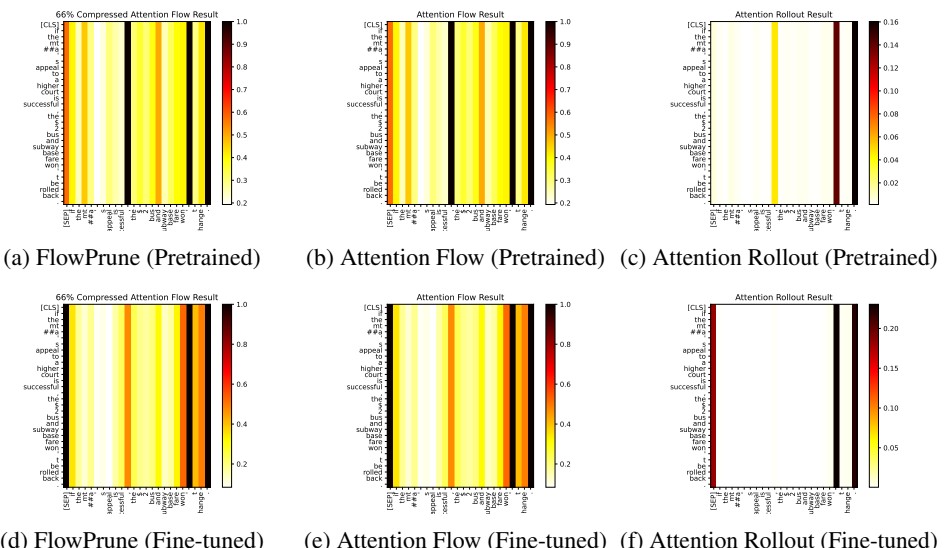

(a) FlowPrune (Pretrained)    (b) Attention Flow (Pretrained)   (c) Attention Rollout (Pretrained)

(d) FlowPrune (Fine-tuned)    (e) Attention Flow (Fine-tuned)   (f) Attention Rollout (Fine-tuned)

Figure 13: **Paraphrasing Verification Example 5.** The two input sentences are: *(1) "If the MTA 's appeal to a higher court is successful , the $ 2 bus and subway base fare won 't be rolled back ."* and *(2) "If the MTA 's appeal is successful , the $ 2 bus and subway base fare won 't change ."*

## A.2 Vision Transformer (ViT) Heatmaps

To further evaluate the utility of FlowPrune in visual interpretability tasks, we provide additional examples using the Vision Transformer (ViT) model. Specifically, we apply FlowPrune to the **DeiT-Small**[41] model on the **ILSVRC2012**[42] image classification dataset. The aim is to generate attention heatmaps that reflect the relevance between each input image token and the *[CLS]* token, which is responsible for final classification decisions. In this setting, we compare three interpretability methods: FlowPrune, the original Attention Flow, and Attention Rollout.

Figures 14 through 18 illustrate five representative examples of attention heatmaps produced by these three methods. As shown in the visualizations, FlowPrune yields results that closely match those of the full Attention Flow, successfully highlighting semantically relevant regions in the image. In contrast, Attention Rollout generates more diffuse and less informative maps, often failing to localize key visual areas.

These results demonstrate that FlowPrune not only retains the interpretability of the original attention mechanism but also achieves substantial computational savings, making it a practical and efficient alternative for ViT visualization.

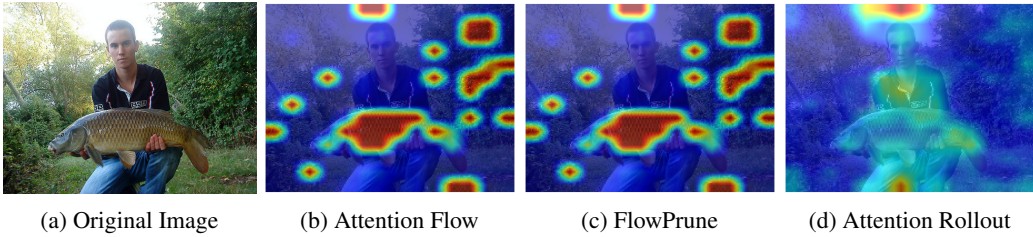

|     |     |     |     |
| --- | --- | --- | --- |
| (a) Original Image | (b) Attention Flow | (c) FlowPrune | (d) Attention Rollout |

Figure 14: **ViT Heapmap Example 1.**

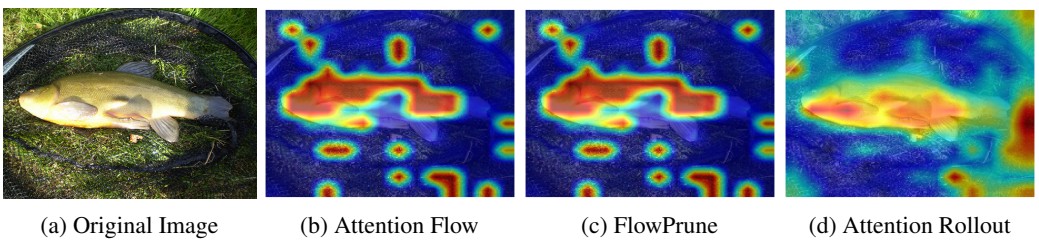

|     |     |     |     |
| --- | --- | --- | --- |
| (a) Original Image | (b) Attention Flow | (c) FlowPrune | (d) Attention Rollout |

Figure 15: **ViT Heapmap Example 2.**

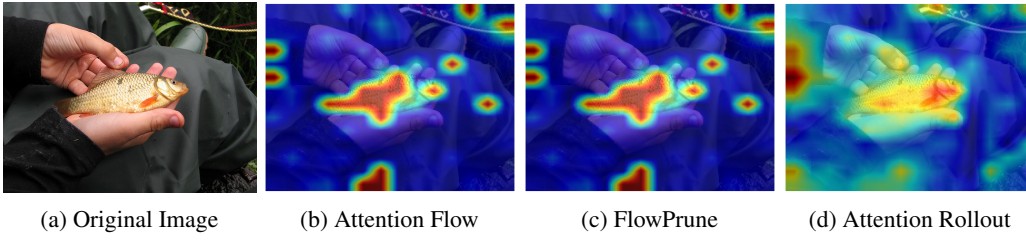

|     |     |     |     |
| --- | --- | --- | --- |
| (a) Original Image | (b) Attention Flow | (c) FlowPrune | (d) Attention Rollout |

Figure 16: **ViT Heapmap Example 3.**

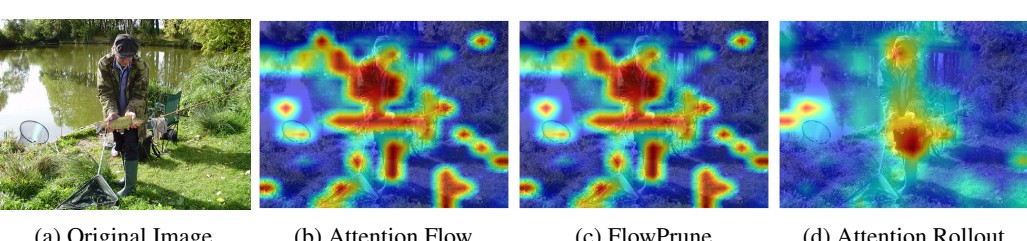

|     |     |     |     |
| --- | --- | --- | --- |
| (a) Original Image | (b) Attention Flow | (c) FlowPrune | (d) Attention Rollout |

Figure 17: **ViT Heapmap Example 4.**

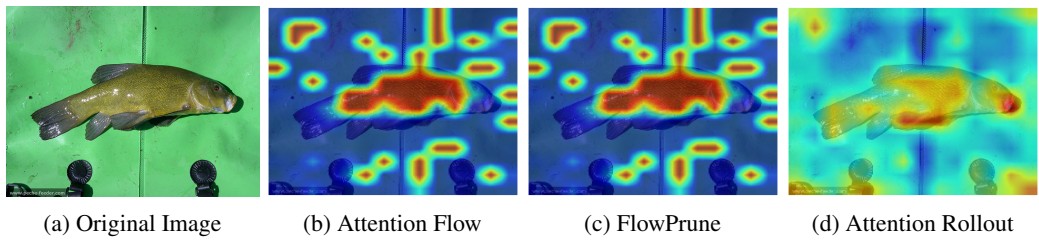

|              |                 |              |                    |
|:------------:|:---------------:|:------------:|:------------------:|
| (a) Original Image | (b) Attention Flow | (c) FlowPrune | (d) Attention Rollout |

Figure 18: **ViT Heapmap Example 5.**

### A.3 Question Answer Verification

The Question Answer Verification (QAV) experiment aims to investigate how fine-tuning affects BERT's ability to validate whether a given answer correctly addresses a question. Using the QNLI [40] dataset from GLUE, the study fine-tunes a BERT-base model and employs Attention Flows to analyze changes in the model's attention mechanisms. The results show that fine-tuning enables the model to better focus on task-relevant details, such as key connecting words and phrases between the question and answer, thereby improving its accuracy in determining the validity of the answer.

In this experiment, we continue to utilize a 12-layer BERT model. We aim to compare the outcomes of FlowPrune and Attention Rollout with those of the original Attention Flow, both in pre-trained and fine-tuned models. The primary focus of this task is the attention metrics from tokens in the initial layer that point to the *[cls]* token in the final layer. Tokens with the highest computed attention metrics are deemed the most significant.

Table 3: **Max-$n$ Overlap for Question Answer Verification.** FP represents Flowprune and AR represents Attention Rollout.(Mean $\pm$ SE)

| Max-$n$ | FP Pretrained | AR Pretrained | FP Fine-tuned | AR Fine-tuned |
|:-------:|:-------------:|:-------------:|:-------------:|:-------------:|
| $n = 3$  | $98.05 \pm 1.11\%$ | $64.68 \pm 2.91\%$ | $96.10 \pm 1.54\%$ | $85.71 \pm 2.59\%$ |
| $n = 5$  | $97.31 \pm 1.54\%$ | $84.48 \pm 2.14\%$ | $94.16 \pm 2.31\%$ | $76.31 \pm 2.21\%$ |
| $n = 10$ | $97.01 \pm 1.72\%$ | $66.50 \pm 1.88\%$ | $93.42 \pm 2.60\%$ | $71.03 \pm 1.76\%$ |

|                          |                              |                                |
|:------------------------:|:----------------------------:|:------------------------------:|
| (a) FlowPrune (Pretrained) | (b) Attention Flow (Pretrained) | (c) Attention Rollout (Pretrained) |
| (d) FlowPrune (Fine-tuned) | (e) Attention Flow (Fine-tuned) | (f) Attention Rollout (Fine-tuned) |

Figure 19: **Question Answer Verification.** Visualization of attention flow results across three methods—FlowPrune, original Attention Flow, and Attention Rollout.

We quantify the **Max-n Overlap** between the most salient tokens derived via FlowPrune/Attention Rollout and the original Attention Flow. For FlowPrune, a retain rate of $67\%$ is employed, thereby

condensing the original 12-layer attention map into 8 layers. Analogous to the Top-$K$ Token Retention method, the Jaccard overlap (IOU) is utilized as the evaluation metric. The results are presented in Table 3. Additionally, an illustrative example from the MPRC dataset is depicted in Figure 19.

## A.4  Layer Selection via Dynamic Programming for FlowPrune

To identify which attention layers to compress in the FlowPrune framework, we formulate the selection process as a dynamic programming (DP) problem that aims to **preserve critical regions in the attention flow graph** while reducing computational cost.

As described in the main text, the attention structure of Transformers forms a strictly layered directed acyclic graph (DAG), in which each token in layer $i$ attends to all tokens in layer $i + 1$. This layered property enables layer-wise flow analysis and facilitates compression strategies based on edge removal and reconnection.

To estimate the relative importance of each layer, we apply a sampling-based heuristic inspired by the Max-Flow Min-Cut Theorem. Specifically, we sample multiple source-to-sink paths through the attention graph and treat the lowest-capacity edge on each path as a *potential min-cut edge*. For each intermediate layer $i$, we count the number $a_i$ of such edges it contains. A lower count indicates that the layer contributes less to the overall minimum cut and is thus a better candidate for compression.

Given these layer-wise scores $\{a_1, a_2, \ldots, a_{L-2}\}$, where the first and last layers are excluded from compression for structural consistency, our goal is to select $N$ layers to retain (i.e., not compress) such that the total number of potential min-cut edges in the retained layers is maximized.

We define the dynamic programming state as:

- $f(i, j, 0)$: the maximum total number of potential min-cut edges when the first $i$ layers are reduced to $j$ layers, and the $i$-th layer is compressed.
- $f(i, j, 1)$: the same, but the $i$-th layer is retained.

The recurrence relations are:

$$f(i, j, 0) = \max\left(f(i - 1, j, 0),\ f(i - 2, j - 1, 1)\right)$$
$$f(i, j, 1) = \max\left(f(i - 1, j - 1, 0),\ f(i - 1, j - 1, 1)\right) + a_i$$

The boundary conditions are:

$$f(0, 0, 0) = -\infty, \quad f(0, 0, 1) = 0$$
$$f(i, j, 0) = -\infty \quad \text{if } i \leq j$$
$$f(i, j, 1) = -\infty \quad \text{if } i < j$$
$$f(i, 1, 0) = 0 \quad \text{for } i > 1$$

The optimal value is given by:

$$\max\left(f(L, N, 0),\ f(L, N, 1)\right)$$

where $L$ is the total number of intermediate layers (excluding the first and last).

To reconstruct the compression scheme, we backtrack from the optimal DP state and trace the transition path:

- If the current state is $(l, n, 0)$, we move to the maximum of $f(l-1, n, 0)$ or $f(l-2, n-1, 1)$, and record that layer $l$ is compressed.
- If the current state is $(l, n, 1)$, we move to the maximum of $f(l-1, n-1, 0)$ or $f(l-1, n-1, 1)$, and record that layer $l$ is retained.

This dynamic programming approach ensures that layers essential to maintaining attention flow—those with higher concentrations of potential min-cut edges—are preserved. Meanwhile, layers with relatively minor contributions are removed.

