# OpenReview forum: "FlowPrune: Accelerating Attention Flow Calculation by Pruning Flow Network"
_NeurIPS.cc/2025/Conference — NeurIPS 2025 poster_

### Official Review · Reviewer_qaDf · 2025-07-01

**Clarity:** 3
**Significance:** 2
**Originality:** 3
**Rating:** 4
**Confidence:** 4

**Summary:**

The paper proposes FlowPrune, an efficient variant of the original Attention Flow interpretability technique. Attention Flow is a method for analyzing how information propagates through all layers of a Transformer model by modeling the entire network as a directed graph. Attention Flow has $O(L^{2.5}N^3)$ complexity for $L$ layers and $N$ tokens, making it impractical for large models. To address this, the authors propose FlowPrune, with two modifications over the original method: Edge Pruning and Layer Compression, both designed to reduce the attention graph while preserving its critical flow structure, giving a complexity of $O(L^{2.5}N^3/R^{2.5})$.

The authors evaluate FlowPrune on LLaMA and LLaVA across multiple compression rates, measuring time speedup, approximation accuracy, and practical utility in real-world scenarios including paraphrasing verification and Vision Transformer heatmap generation.

**Questions:**

Does the upper-bound method, Attention Flow, perform comparatively to other established input attribution techniques?

It is counterintuitive that higher compression rates, close to 1, reflect more layers retained. Typically, I would associate higher compression rates with keeping less number of layers.

l218: "L/R layers reduces" -> "R/L layers reduces"

Typos:

l29: propagateed -> "propagated"

Inconsistent use of "LLama" and "Llama"

**Ethical Concerns:**

["NO or VERY MINOR ethics concerns only"]

**Final Justification:**

The authors have provided useful clarifications and additional evaluations that strengthen the work. However, the main concern remains, the paper does not sufficiently justify the choice why attention flow is preferable to established attribution methods. Thus, a rating of 4 is appropriate.

**Limitations:**

yes

**Quality:**

3

**Strengths And Weaknesses:**

Strengths

- The two proposed techniques, edge pruning and layer compression, are theoretically sound and well-motivated by the structural properties of Transformer architectures.

- The evaluation uses well-designed metrics for comparing against ground-truth attention flow, demonstrating that FlowPrune maintains high fidelity to the original method even at aggressive compression rates (few number of layers retained in the attention graph).

Weaknesses

- The paper does not justify why attention flow is preferable to established attribution methods like gradients or integrated gradients. The original approximation method, "Attention Rollout" ([Abnar et al., 2020](https://aclanthology.org/2020.acl-main.385/)) has been shown to underperform compared to alternative methods ([Chefer et al., 2021](https://openaccess.thecvf.com/content/CVPR2021/papers/Chefer_Transformer_Interpretability_Beyond_Attention_Visualization_CVPR_2021_paper.pdf), [Mohebbi et al., 2023](https://aclanthology.org/2023.eacl-main.245/)). A key limitation of attention flow, and by extension FlowPrune, is its inability to provide class-specific explanations, which restricts its applicability. Without a compelling rationale for using attention flow, accelerating this particular method offers limited practical value.

- The layer compression component removes intermediate layers, undermining the method's ability to analyze actual information flow through the original network architecture.

- Results are showed depending on layer compression rate, but not with respect to edge pruning threshold. You say the edge pruning threshold is set to 1e−6 throughout all experiments. How did you choose this threshold? What is the impact of this threshold in the metrics?

- The evaluation is limited to two models (LLaMA, LLaVA), broader evaluation across architectures would strengthen claims.

---

> ### Author Rebuttal · Authors · 2025-07-30
>
> Dear reviewer:
>
> Thank you for your constructive comments, Your valuable comments have been of considerable assistance to us.
>
> First, we would like to emphasize the novelty of our FlowPurne method. This method helps to reduce the running time for interpreting behaviors of large-scale transformer models. Multiple well-designed metrics are used to evaluate the efficiency and interpretation quality.
>
> Then, to your concerns, we address the comments as follows:
>
> **1. Attention flow as interpretability tool.** In [14], Attention Flow is defined by modeling the multi-layer self-attention mechanism of the Transformer as a capacitated flow network, where attention weights and residual connections jointly determine edge capacities. The maximum flow between tokens of interest serves as the interpretability metric, quantifying how much information is transmitted between tokens. This formulation has been adopted in works such as [22] to analyze and compare attention mechanisms. However, computing attention flow is computationally expensive, prompting many researchers to use Attention Rollout as a faster alternative.
>
> However, as you have pointed out, attention rollout often performs poorly. This is because it is a heuristic approximation that relies on direct multiplication of attention matrices, which causes the resulting values to deviate from true attention flow. Actually, this is why we propose *FlowPrune*, which is a method grounded in the Max-Flow Min-Cut theorem that can significantly accelerates attention flow computation while providing more accurate approximations of the true flow.
>
> **2. Class-specific Explanations.** For class-specific explanations, our experiments on BERT (lines 276–301, Table 1) were specifically designed to evaluate the effectiveness of FlowPrune in such contexts. These experiments aimed to identify which tokens are most influential in classification decisions. FlowPrune achieved strong performance on this task, significantly outperforming Attention Rollout. Additionally, as demonstrated in Figure 8 and Figures 6–10 of the Appendix, FlowPrune can effectively highlight the visual regions contributing to the final predicted labels in image classification as the original attention flow, while attention rollout fails . These results support the conclusion that FlowPrune provides meaningful class-specific explanations.
>
> **3. Layer Compression.** As discussed in lines 124–129 and 150–158, by leveraging the Max-Flow Min-Cut Theorem alongside the Transformer’s structural properties, we remove layers that have minimal impact on information flow. This selective pruning aligns with our goal of eliminating layers that contribute little to the results.
>
> **4. Edge Pruning Threshold.** In fact, we have conducted experiments on the selection of the edge pruning threshold, but due to page limitations, these results were not included in the paper. The primary reason for omitting the results on the edge pruning threshold is that the experimental outcomes demonstrate that FlowPrune is not particularly sensitive to its value. Among the tested values, $1 \times 10^{-6}$ consistently performs well and offers a reasonable balance. The table below illustrates the average absolute error for the LLaMA model under different edge pruning thresholds after retaining half of the layers (16 layers).
>
> |1e-8|1e-7|1e-6|1e-5|
> |---|---|---|---|
> |0.0137|0.0099|0.0097|0.0103|
>
> **5. Generalizability.** To evaluate the generalizability of FlowPrune, we have selected two representative large models: the unimodal LLaMA and the multimodal LLaVA. We also assessed its effectiveness on smaller-scale models—BERT for NLP (Sec 4.2.1) and ViT for vision (Sec 4.2.2). In addition to measuring absolute error against the true attention flow (Fig 4), we considered that attention flow is often used to study relative ranking; thus, we introduced corresponding metrics for this purpose (Fig 5, 6). Together, these diverse models and evaluation criteria support the general applicability of our method. In response to your suggestion, we also included experiments on two widely used large models, Qwen (unimodal) and Qwen-VL (multimodal). The following table presents the absolute and relative errors in the high-flow regions, which are of the most interest to researchers, for the Qwen and Qwen-VL models, where we can find that the absolute/relative error is not large.
>
>   | Region\Model| qwen (absolute error) | qwen (relative error) | qwen-vl (absolute error) | qwen-vl (relative error) |
> | --- | --- | --- | --- | --- |
> [0.9-1.0] | 0.003 | 0.31% | 0.001 | 0.12% |
> [0.8-0.9] | 0.014 | 1.63% | 0.012 | 1.41% |
> [0.7-0.8] | 0.017 | 2.28% | 0.013 | 1.71% |
>
> In addition to the aforementioned experiments, we also included an application scenario experiment on Question Answer Verification, which is also based on BERT. You can find the detailed introduction of the experiment in [22]. We compared the Top-K Token Retention with Attention Flow of FlowPrune and Attention Rollout, and organized the results in a manner similar to *Table 1* in Section 4.2.1. The results show that, compared to Attention Rollout, the FlowPrune method is much closer to the analysis results of Attention Flow, demonstrating FlowPrune outperforms Attention Rollout. The table below presents the results:
>
> |Max-n|FP Pretrained(Mean ± SE)|AR Pretrained(Mean ± SE)|FP Fine-tuned(Mean ± SE)|AR Fine-tuned(Mean ± SE)|
> |---|---|---|---|---|
> | n = 3 | 0.9805 ± 0.0111 | 0.6468 ± 0.0291 | 0.9610 ± 0.0154 | 0.8571 ± 0.0259 |
> | n = 5 | 0.9731 ± 0.0154 | 0.8448 ± 0.0214 | 0.9416 ± 0.0231 | 0.7631 ± 0.0221 |
> | n = 10 | 0.9701 ± 0.0172 | 0.6650 ± 0.0188 | 0.9342 ± 0.0260 | 0.7103 ± 0.0176 |
>
> **6. Input Attribution Techniques.** Your question is indeed very insightful. In fact, there has been some debate over the merits of attention-based versus gradient-based attribution methods about 5-6 years ago [A, B]. Wiegreffe and Pinter [B] argued that attention can be "an explanation, not the explanation." This implies that for the same prediction, there may be multiple equally valid and reasonable explanations. We concur with this conclusion and believe that both types of methods have their own strengths.
>
> Researchers generally view these two types of methods as representing different categories of explanations. Attention-based methods focus on Intrinsic Interpretability, utilizing the inherent architectural characteristics of the model to provide explanations. They offer a built-in window into the model's internal "focus" [C]. On the other hand, gradient-based attribution methods focus on Post-Hoc Explanations, generating explanations by probing the model's behavior. This category is primarily dominated by saliency and attribution techniques.
>
> Also, attention-based methods have the advantage of requiring only forward propagation, making them simpler to use and more intuitive [D]. In contrast, gradient-based methods necessitate backward propagation, which demands additional GPU computation and is not supported in some scenarios. For instance, many models undergo quantization after training, and quantized models do not support backward propagation. In such cases, attention-based methods are more convenient. Moreover, for attention flow[14], it has about 1000 citations in Google Scholar, suggesting it has considerable influence and impact in the community.
>
> [A] Attention is not Explanation
>
> [B] Attention is not not Explanation
>
> [C] The Role of Attention Mechanisms in Enhancing Transparency and Interpretability of Neural Network Models in Explainable AI
>
> [D] A Comprehensive Guide to Explainable AI (XAI) for Generative Models
>
> **7.  Compression Rate.** Thank you for pointing out this issue. In our paper, the compression rate is defined as the ratio of the number of **retained layers** to the total number of layers. Therefore, when the compression rate approaches 1, it implies that the number of retained layers is close to the total number of layers, meaning that more layers are preserved. This name is indeed counterintuitive. Following your suggestion, we will rename the compression rate to **retain rate** to align the terminology with its definition. We will correct this name in the revision.
>
> **8. L/R Layers.** In fact, in the context of our paper, it should be $L/R$ layers in line 218. The context means that retaining only $L/R$ layers will reduces the computational time, where $L$ denotes the total number of layers in the model, and $1/R$ represents the compression rate, or retain rate in the revision.

---

> ### Author Response · Authors · 2025-08-06
>
> Dear Reviewers:
>
> We hope this message finds you well.
>
> Thank you again for your valuable feedback on our submission. We have submitted our rebuttal and are eager to engage in a discussion before the phase concludes.
>
> We truly appreciate the time and effort you have dedicated to reviewing our paper. We believe the points you raised are crucial for improving our work and are eager to provide any further clarifications that may be needed.
>
> As the discussion phase is approaching its end, we would greatly appreciate the opportunity to provide any further clarifications and address any remaining questions you may have.
>
> Thank you for your time and dedication to the review process.
>
> Sincerely,
>
> Authors of Submission 18977

---

> ### Comment · Reviewer_qaDf · 2025-08-07
> **Response to Rebuttal**
>
> I thank the authors for the detailed response. I appreciate the clarifications and additional experimental evidence on Qwen.
>
> > Attention flow as interpretability tool // Input Attribution Techniques.
>
> I understand that Attention Flow is a more faithful method than Attention Rollout, which is an approximation. However, simplifying or accelerating a method only has practical value if there is a clear justification for why that method is worth using in the first place. While FlowPrune is evaluated based on its agreement with Attention Flow, particularly in terms of top-k token overlap (e.g., via IoU), this evaluation assumes that those top-k tokens are meaningfully relevant.
>
> In the broader literature on input attribution, faithfulness is typically assessed through metrics like comprehensiveness and sufficiency (e.g., DeYoung et al., 2020), which test whether the identified tokens actually influence the model's predictions. Without such analysis, it's unclear whether the explanations provided by FlowPrune, despite matching Attention Flow, are actually useful or reliable.
>
> De Young et al., 2020: Eraser: A Benchmark to Evaluate Rationalized NLP Models. _Proceedings of the 58th Annual Meeting of the Association for Computational Linguistics_.
>
> > Class-specific Explanations.
>
> What I was referring to is the ability to generate class-specific explanations, that is, explanations not only for the predicted class but also for alternative classes. Gradient-based methods, for example, allow computing gradients with respect to each class output, making it possible to identify which input tokens contributed to the prediction, as well as to the other class outcomes.

---

> ### Author Response · Authors · 2025-08-08
>
> Thank you for your valuable feedback. We appreciate the opportunity to further clarify our methodology and the rationale behind our use of attention flow.
>
> Actually, the paraphrasing verification experiment in **Section 4.2.1** performs a similar function to the analysis of **Agreement with Human Rationales** in DeYoung et al., 2020. Our approach, which analyzes the **Attention Flow Value to the [CLS] token**, serves a comparable purpose by identifying the key tokens the model prioritizes for its decision.
>
> We agree with your assessment that directly using attention flow to analyze contributions to different labels presents a challenge. However, we maintain that attention flow is a powerful tool for understanding model behavior, particularly from an information-flow perspective. Our purpose was to explore these information pathways, not to perform a traditional feature importance analysis. We'd like to share a few examples from recent literature to support this point:
>
> * A recent **EMNLP Best Paper** [A] successfully used attention flow to analyze how LLMs process information during in-context learning. This work shows that by studying how information flows through the model's layers, we can gain valuable insights into its reasoning process, even without directly quantifying each token's contribution to a specific label.
> * The core objective of [B] is to visualize the "attention flow" within the ViT network for the first time using the proposed DAAM method, in order to explain how decisions evolve. The paper indicates that attention flow can help qualitatively analyze how modules inserted into the ViT model (such as XCA or T2T modules) influence the formation of object attention.
> * This paper [C] used attention flow for multi-scale analysis, noting that through a focused token-level view, users can understand how attention flows between specific parts of a document.
>
>
> We hope these examples provide a clearer understanding of our experimental design and address your concerns.
>
>
> [A] Label Words are Anchors: An Information Flow Perspective for Understanding In-context Learning
>
> [B] Dynamic Accumulated Attention Map for Interpreting Evolution of Decision-Making in Vision Transformer
>
> [C] A Sentence-Level Visualization of Attention in Large Language Models

---

### Official Review · Reviewer_Ttwq · 2025-07-02

**Clarity:** 2
**Significance:** 3
**Originality:** 3
**Rating:** 5
**Confidence:** 4

**Summary:**

This paper proposes FlowPrune, a method to accelerate the computation of attention flow, an interpretability method in NLP and vision which tries to incorporate the attention information from all layers of the model when computing interpretable heatmaps over the inputs. The way FlowPrune achieves speedups is twofold: (1) it prunes edges in the attention graph for which the attention weights fall below a certain threshold and (2) it removes layers of the Transformer for which the edges contained in the layer are unlikely (computed through some heuristic) to contain any of the min-cut edges. The authors test for speedup and accuracy on the LLama and Llava models, and use BERT and ViT for the application domains. Overall, both quantitatively and qualitatively, they find FlowPrune to match the full attention flow results pretty well.

**Questions:**

**Clarifications**
1. Line 193: "... we provided 1,000 inputs ..." -> "inputs" means questions from the dataset here?
2. Line 212: does "samples" mean questions from the dataset here?
3. Figure 5: are these plots averages over the 1000 samples drawn from the dataset? Because if not, how can the top3 be in the 90s? It should only be able to be 0, 1/3, 2/3, or 1.

**Grammar**
1. Line 29: change "propagateed" to "propagated"
2. Line 136: change "In Transformer, softmax operation" to "In Transformers, the softmax operation"
3. Caption of Figure 1: change "pathes" to "paths" and change "brwon" to "brown"
4. Line 162 - 163: "However, in more realistic ... this assumption less precise." -> this sentence doesn't read quite right grammatically, maybe rephrase.
5. Line 227: change "may retains" to "may retain"?
6. Line 306: change "expriments" to "experiments"

**Ethical Concerns:**

["NO or VERY MINOR ethics concerns only"]

**Final Justification:**

The weaknesses brought up in my initial review have been sufficiently addressed, and hence I have increased my score to 5.

The main key issue that was addressed was that I was initially not convinced the paper was really using any property from the Max-Flow Min-Cut Theorem, but the authors explained that lines 127 - 130 are really a corollary, not just an intuition. I would encourage the authors to include this explanation/proof into the next revision of their paper.

**Limitations:**

yes

**Quality:**

3

**Strengths And Weaknesses:**

## Strengths
### Quality
1. The quantitative results in Table 1 and 2 are very convincing: they clearly demonstrate FlowPrune is a solid improvement over Attention Rollout compared with the original attention flow. Qualitatively, the heatmaps in Figure 7 and 8 also look quite convincing.
2. The metrics presented in Section 4.1.1 are well-designed to measure the relevant factors of their method (speedup + error analysis).
3. Line 247 - 251: I like that the authors connect the low approximation errors for high attention flow values to what actually matters in practice, which are the top source-destination pairs with the highest attention flow values.

### Clarity
1. Generally, I find the paper to have clear graphs and tables that are easy to read and clearly convey the results.

### Originality
1. I generally like the authors' idea of using insights from graph theory to improve computational efficiency, and the authors' use of max-flow min-cut seems pretty original to me!

### Significance
1. Given the strong results, I imagine this could become the current default method when computing attention flow which seems impactful.

## Weaknesses
### Quality
1. Line 163 - 165: "Nevertheless, attention graphs ... to all tokens in layer i + 1". Two things: (1) typical Transformer architectures (e.g. Llama) are NOT fully connected between layers since they use causal attention masks, and (2) shouldn't this be from i + 1 to i? (1) pointed out above also draws into the question the line that follows: "This near-uniform connectivity implies that sampled paths are likely to overlap on any edge." How are GPT-style models, which don't have near-uniform connectivity, affected here?
2. Line 166 - 168: "As a result, the edge ... more likely to lie on the global min-cut, ..." -> Is this true? Do the authors have evidence or proof for this?
3. Line 219: "The empirical results in Figures 2 and 3 generally align with this analysis." -> Is this really the case? It doesn't seem to me the curves in Figure 2 are really following the predicted $1/R^{2.5}$ speedup prediction? Maybe the authors could include the theoretical prediction in Figure 2 so this is easier to compare? The authors also state that "LLaVA's performance closely matches the theoretical expectation" but I still don't agree based on the $1/R^{2.5}$ factor.
4. Line 243 - 246: "(2) In Layer Compression, newly added edge ... not decreases" -> Why do we set them all to one? This seems to break the constraint on attention weights that they should sum to one? And also the last part says that "this ensures that the maximum flow from the compressed graph increases, not decreases", is this desirable? Or why is this not a bad thing? Ideally I thought we want the max flow to stay roughly the same, not increase.
5. Is it possible to add standard errors to Table 1 and 2?
6. The authors discuss two techniques for reducing the attention graph: (1) edge pruning and (2) layer reduction. However, it seems there is no experiments specifically investigating (1)?

### Clarity
1. Line 57 - 58: "due to the softmax operation, many attention weights are close to zero ..." -> I think this is stated a few times in the paper, but it's never really made clear *why* this is due to the softmax operation. Could the authors provide a short explanation for this? Is this due to the exponential in the softmax?
2. Line 127 - 130: "This theory provides ... without minimum cut." -> Why is this true? Or is this just an intuition? Can't removing stuff affect what constitutes the minimum cut?
3. Figure 1(b): why are the orange lines here the min-cut edges? And which is the source and sink in this diagram?
4. Line 235: "... overhead becomes negligible after computing more than 6 casees for Llama and 2 cases for LLaVA." How did the authors get these numbers? Can you help me understand how to get these from the graphs?
5. Line 266 - 268: "This result is fully sufficient for practical applications with lower precision requirements, whilte the results at a compression rate 50% are adequate for tasks with higher precision demands." -> What does "precision requirements" mean here? And how do we know its fully sufficient?

### Significance
1. Line 120-121: "this high cost makes the method impractical for large-scale Transformer models" -> Could the authors shed some light on how computationally expensive this would actually be for some large models? For example, in wall clock time, how long woudl it take to compute the full heatmap for a single example say? I listed this under significance since the answer to this question can have some impact to how significant the speedups really are, and hence what impact the method will have in the field.

---

> ### Author Rebuttal · Authors · 2025-07-30
>
> Dear reviewer:
>
> Thank you for your constructive comments, Your valuable comments have been of considerable assistance to us. Then, to your concerns, we address the comments as follows:
>
> **1. Attention Flow is calculated after token-by-token generation instead of during generation.** In fact, when using attention flow for analysis, the entire sequence is input into the Transformer **in a single parallel forward pass** to obtain the corresponding attention graph. Consequently, the causal attention mask used in autoregressive generation is not applied here. Furthermore, since attention flow tracks information propagation from the first to the last layer, the attention graph is formed by edges connecting layer $i$ to layer $i+1$, reflecting the forward flow of information within the model.
>
> **2. Question about Line 166 - 168.** As noted in lines 159–168, this is a heuristic conjecture for which we currently lack a formal proof. Nevertheless, both our fundamental evaluations and experiments support its validity. Specifically, the experiments on LLaMA and LLaVA in Section 4.1 show that FlowPrune have low approximation errors, suggesting that the proposed approach is empirically reliable.
>
> **3. Bound in Lines 216-223.** The bound presented in lines 216–223 serves as an estimate, based on the assumption that when the proportions of pruned edges and nodes in FlowPrune are approximately equal, the bound reasonably approximates the actual speedup. Additionally, the specific structure of the attention graph can influence the extent of acceleration achieved. We will incorporate the above explaination in revision.
>
> In lines 224–228, we briefly analyze possible reasons why the LLaMA curve deviates from the theoretical prediction. We attribute this discrepancy to differences in input token types across models. Unlike LLaMA, LLaVA is a multimodal vision-language model that processes a large number of visual tokens, many of which are redundant. This redundancy results in a higher proportion of near-zero attention weights, leading to more edges being pruned, making this case more similar to the assumption. As a result, the compressed attention graph in LLaVA aligns more closely with our theoretical assumptions.
>
> **4. Setting edge capacity to 1.** According to the Max-Flow Min-Cut Theorem, increasing the capacity of edges **not in the minimum cut** does not affect the maximum flow value, as the flow is bounded by the capacity of the minimum cut. In FlowPrune, we ensure that the removed parts contain as few minimum cut edges as possible.  To maintain the maximum flow value after compression, newly added shortcut edges are assigned sufficiently large capacities to ensure that they do not become part of the new minimum cut. In our case, a capacity of **1** is considered sufficiently large.
>
> Note that removal involves deleting certain vertices along with their connected edges. For any pair of remaining vertices that were previously connected via a path through the removed nodes, we introduce a new edge with large capacity to restore connectivity. Here, a capacity of 1 is adequate because we do not compress the first attention layer. As a result, the total flow emitted from the source remains bounded by 1, meaning the total maximum flow will also not exceed 1. Consequently, added edges with capacity 1 will not appear in the minimum cut. While this breaks the constraint that attention weights should sum to one, preserving the first layer ensures the attention flow remains within [0, 1], maintaining its interpretability.
>
> **5. Increasing Max-Flow is Better than Decreasing.** Since the maximum flow is bounded above by 1 in both the original and compressed attention graphs, a relatively large original flow value restricts the possible outcomes of FlowPrune to a narrower range, thereby reducing potential error. As discussed in lines 247–251, such cases are considered important, and thus, smaller errors are particularly desirable.
>
> **6. Add Standard Errors.** Thank you for your suggestion. We will add the following results about standard errors in revision.
>
> |Max-n|FP Pretrained(Mean ± SE)|AR Pretrained(Mean ± SE)|FP Fine-tuned(Mean ± SE)|AR Fine-tuned(Mean ± SE)|
> |---|---|---|---|---|
> |n = 3|0.9850 ± 0.0086|0.6340 ± 0.0258|0.9500 ± 0.0150|0.8450 ± 0.0232|
> |n = 5|0.9793 ± 0.0119|0.8471 ± 0.0183|0.9250 ± 0.0226|0.7662 ± 0.0193|
> |n = 10|0.9769 ± 0.0133|0.6638 ± 0.0158|0.9145 ± 0.0258|0.7085 ± 0.0156|
>
> |Max-n|FP(Mean ± SE)|AR(Mean ± SE)|
> |---|---|---|
> |n = 10|1.0000 ± 0.0000|0.2818 ± 0.0306|
> |n = 20|0.9953 ± 0.0012|0.5084 ± 0.0374|
> |n = 30|0.9959 ± 0.0011|0.7852 ± 0.0343|
>
> **7. Edge Pruning Experiment.**  Please see ``**1.Static Threshold.**'' in response to Reviewer b9Ta for more details.
>
> **8. Small Attention Weights.** Your observation is accurate. Due to the exponential nature of the softmax function, only a few    attention values are significantly greater than zero. For example, in our experiments across the four models used in this study (LLaMA, LLaVA, BERT, and ViT), more than 95% of the attention values are smaller than 1 \times 10^{-6}.
>
> **9. Intuition in Lines 127 - 130.** We acknowledge that the term we used was imprecise; what we intended to convey is more accurately described as a **corollary**, rather than an intuition. Specifically, we refer to the removal of certain vertices from the graph, along with their incident edges. For the remaining vertices, if any pair was previously connected via a path composed of the removed edges, we introduce a new edge with sufficiently large capacity to preserve connectivity. Under this construction, if none of the removed edges belong to the original minimum cut, the minimum cut remains unchanged. This holds because, after the removal, any S–T cut must either  1) includes the newly introduced edges and, due to their sufficiently large capacity, cannot be the minimum cut, or 2) is composed entirely of original edges and thus cannot be smaller than the minimum cut of the original graph.
>
> Since the minimum cut of the original graph indeed exists within the graph, it remains the minimum cut of the graph after the removal operation, thereby preserving the invariance of the maximum flow.
>
> **10. Question About Figure 1(b).** We apologize for the confusion. The figure illustrates our sampling-based heuristic algorithm. The orange lines indicate *potential* min-cut edges, not actual min-cut edges. This algorithm is introduced in lines 153–158, with its motivation discussed in lines 159–168. Note that the figure does not include predefined source or sink nodes; these are determined only when computing the attention flow between a specific pair of tokens.
>
> **11. Overhead of Flowprune.** For a single attention graph, pruning is performed only once, after which the sparsified graph is reused to compute attention flow between all token pairs. Although pruning introduces initial computational overhead, this cost is amortized as the number of evaluated token pairs increases. For example, in the case of LLaMA, computing attention flow for 6 token pairs on the original attention graph takes approximately the same time as applying FlowPrune once and then performing max-flow computations on the sparsified graph for those pairs. As more pairs are calculated, the average time per pair for FlowPrune continues to decrease. This trend is shown in Figure 3, where we plot average per-pair computation time under various compression rates. The intersection with the baseline curve (original attention flow time) indicates the point beyond which FlowPrune offers a computational advantage.
>
> **12. Precision Requirements.** As discussed in lines 68–70, attention flow is often used to assess the relative importance of tokens based on the ranking of flow values. Our notion of precision pertains specifically to this context. At a 50% compression rate, the first instance of incorrect relative ordering appears only after the top 40% of token pairs. In practice, applications that rely on attention flow typically focus on the top 5–10 token pairs. Therefore, an FDR of 0.4 is sufficient for these use cases.
>
> The comma in lines 266-268 you cited, is a typo. The lines you cited should be two separate sentences. The corrected version should be "This result is fully sufficient for practical applications with lower precision requirements. While the results at a compression rate 50% are adequate for tasks with higher precision demands.".
>
> The first sentence discusses the scenario of retaining only 3 layers, while the second sentence addresses the situation with a compression rate of 50%. We apologize for any confusion this may have caused.
>
> **13. Actual Run Time.** The maximum flow algorithm is a classical graph-theoretic method that typically lacks efficient GPU-accelerated implementations. Our experiments were conducted on a server equipped with two AMD EPYC 7453 28-core processors. Using LLaVA as an example, we measured the runtime for computing attention flow across 100 token pairs in a sample containing 400–500 tokens within a 32-layer LLaVA model. With the original attention flow algorithm, the runtime reaches 112 minutes. In contrast, compressing the attention graph to 12 layers reduces the runtime to approximately 10 minutes.
>
> **14. Input and Samples.** We construct input sequences by concatenating questions with their corresponding answers from the dataset. Each input generates a corresponding attention graph, which we refer to as a sample.
>
>  **15.Average Results.** Indeed, that is precisely the case.
>
>  **16.Grammar.** Thanks for pointing out these mistakes and we will follow your suggestion to revise them.

---

> > ### Comment · Reviewer_Ttwq · 2025-08-04
> > **Thank you**
> >
> > I thank the authors for their extensive rebuttal. It has mostly addressed my confusions, and I have therefore increased my score. However, I do have a few remaining questions:
> > 1. > Attention Flow is calculated after token-by-token generation instead of during generation. In fact, when using attention flow for analysis, the entire sequence is input into the Transformer in a single parallel forward pass to obtain the corresponding attention graph. Consequently, the causal attention mask used in autoregressive generation is not applied here.
> >
> > For models that use causal attention masks (e.g. GPT), it seems strange that the Attention Flow would ignore this, as then it would be taking into account the influence of future tokens for past tokens? It would be great if the authors could further clarify this.
> >
> > 2. > The bound presented in lines 216–223 serves as an estimate, based on the assumption that when the proportions of pruned edges and nodes in FlowPrune are approximately equal, the bound reasonably approximates the actual speedup. Additionally, the specific structure of the attention graph can influence the extent of acceleration achieved. We will incorporate the above explaination in revision.
> >
> > Does this mean neither curve in Figure 2 is close to the theoretical prediction? (which is what it seems like to me) If so, please revise this in the next version of the pdf.

---

> ### Author Response · Authors · 2025-08-05
>
> Thank you for your insightful comments, which have helped us further refine our study. Below, we address the two questions raised during the discussion.
>
> **1. Impact of causal attention on attention flow.**
>
> You are correct that causal masking can affect the computation of attention flow. However, we find its practical impact to be limited for two main reasons. First, during supervised fine-tuning (SFT), models are often trained using full-sentence prompts followed by token-by-token generation. In such cases, causal masking is only applied during the generation phase, not when processing the initial full prompt. This setup naturally mitigates the potential distortion introduced by the causal mask. Second, and more importantly, when attention flow is used for interpretability purposes, the analysis typically focuses on how tokens in the earlier part of a sentence influence a predicted token later in the sequence. This design implicitly offsets the impact of causal masking. For example, in the sentence "This movie is amazing, I __ it", suppose the predicted token is "like". To assess how "amazing" contributes to this prediction, attention flow is computed from "amazing" to "like". Although indirect paths such as amazing → it → like may exist, our empirical observations suggest that their influence is relatively minor. As such, causal masking has limited effect on the practical use of attention flow for interpretability.
>
> **2. Gap between theoretical bounds and empirical curves.**
>
> Regarding the curve of LLaVA, it indeed aligns more closely with our theoretical bound compared to LLaMA, though a  gap still remains. We will follow your suggestion to revise this part of the paper to clarify the distinction and avoid potential misunderstandings.
>
> Once again, we sincerely appreciate your questions, which further prompted us to revisit and deepen our analysis. We will incorporate these clarifications into the revision.

---

> ### Comment · Reviewer_Ttwq · 2025-08-05
>
> Thanks for the follow-up.
>
> > First, during supervised fine-tuning (SFT), models are often trained using full-sentence prompts followed by token-by-token generation. In such cases, causal masking is only applied during the generation phase, not when processing the initial full prompt.
>
> I don't think this is correct. For any decoder-style model (e.g. Llama), there will be causal masking (i.e. a token can only ever attend to itself or past tokens) at all times and for all tokens. In addition, there is no token-by-token generation in SFT training, since it's all offline.

---

> > ### Author Response · Authors · 2025-08-09
> >
> > Thank you for this insightful question.
> >
> > In response to your query, we conducted some empirical observations and found that the influence of the causal mask appears to be minimal in practice. We rarely observed information flow propagating through uncausal paths. That is, to transmit flow through tokens that come after the tokens of interest in the sequence.
> >
> > We agree this is a crucial point for understanding the underlying mechanism, and we will continue to explore the impact of the causal mask in our future work.

---

### Official Review · Reviewer_suGC · 2025-07-02

**Clarity:** 3
**Significance:** 2
**Originality:** 3
**Rating:** 3
**Confidence:** 3

**Summary:**

This paper presents FlowPrune, a framework designed to accelerate attention flow computation in Transformers by pruning attention graphs prior to applying max-flow algorithms. The approach integrates two techniques: Edge Pruning, which removes low-capacity attention edges, and Layer Compression, which eliminates layers contributing minimally to the flow. These strategies collectively reduce computational complexity while maintaining interpretability fidelity. The authors evaluate FlowPrune on case studies across both NLP and vision tasks, demonstrating that it yields interpretability results consistent with those of the original Attention Flow method.

**Questions:**

The authors mentioned that "Several approximations, including Attention Rollout [14], have been proposed to reduce complexity, though they often sacrifice theoretical properties such as flow conservation." I’m wondering whether the proposed approach provides any theoretical guarantees on the approximation error.

**Ethical Concerns:**

["NO or VERY MINOR ethics concerns only"]

**Final Justification:**

The response addresses most of my concerns; however, the justification for choosing the max-flow/min-cut approach over alternative solutions remains insufficient, leaving their design choice without adequate motivation.  Also, the claim regarding theoretical guarantees appears overstated. Based on these points, I will maintain my original score.

**Limitations:**

Although some limitations are acknowledged, a more comprehensive discussion would strengthen the paper.

**Quality:**

3

**Strengths And Weaknesses:**

Strength:
- The target problem of accelerating attention flow computation is well-motivated, given its high computational complexity.

- The paper is clearly presented, with good writing quality and logical organization. Additionally, the public release of the code enhances the reproducibility and practical value of the work for the research community.

- To the best of my knowledge, the use of the Max-Flow Min-Cut approach in this context is among the early explorations, making it a novel direction.


Weakness:
- The choice of the Max-Flow Min-Cut approach requires further justification. It would strengthen the paper if the authors could clarify why this method is preferable over other graph sparsification or general sparsification techniques. Including additional ablations comparing against these alternatives would help improve the paper’s rigor.

- The generalizability of the proposed method warrants further evaluation. Currently, the experiments are limited to LLaMA and LLaVA models. Even within this small-scale evaluation, performance patterns vary significantly (e.g., Fig. 4). It remains unclear whether the proposed method consistently performs well across different models and tasks. Broader evaluation on additional benchmarks would be beneficial.

- The paper would benefit from a more thorough discussion of the error distribution in low-attention regions. As shown in Fig. 4, the relative errors are quite high for low attention values. The authors should discuss potential causes, limitations, and possible mitigation strategies for this issue.

---

> ### Author Rebuttal · Authors · 2025-07-30
>
> Dear reviewer:
>
> Thank you for your constructive comments, Your valuable comments have been of considerable assistance to us. Then, to your concerns, we address the comments as follows:
>
>  **1.Other Sparsification Techniques.** We build FlowPrune on the Max-Flow Min-Cut Theorem because attention flow is inherently formulated as a maximum flow problem over the attention graph[14]. As a cornerstone, the Max-Flow Min-Cut Theorem provides the primary theoretical foundation for designing and analyzing max-flow algorithms. To our knowledge, few alternative frameworks offer comparable guidance for optimizing max-flow computations. Accordingly, in developing FlowPrune, we focused on this theorem in conjunction with exploiting the structural properties of Transformer attention graphs, rather than adopting other graph sparsification techniques that are less directly aligned with the max-flow formulation.
>
> **2. Generalizability.** To evaluate the generalizability of FlowPrune, we have selected two representative large models: the unimodal LLaMA and the multimodal LLaVA. We also assessed its effectiveness on smaller-scale models—BERT for NLP (Sec 4.2.1) and ViT for vision (Sec 4.2.2). In addition to measuring absolute error against the true attention flow (Fig 4), we considered that attention flow is often used to study relative ranking; thus, we introduced corresponding metrics for this purpose (Fig 5, 6). Together, these diverse models and evaluation criteria support the general applicability of our method. In response to your suggestion, we also included experiments on two widely used large models, Qwen (unimodal) and Qwen-VL (multimodal). The following table presents the absolute and relative errors in the high attention regions, which are of the most interest to researchers, for the Qwen and Qwen-VL models, where we can find that the absolute/relative error is not large.
>
> | Region\Model| qwen (absolute error) | qwen (relative error) | qwen-vl (absolute error) | qwen-vl (relative error) |
> | --- | --- | --- | --- | --- |
> [0.9-1.0] | 0.003 | 0.31% | 0.001 | 0.12% |
> [0.8-0.9] | 0.014 | 1.63% | 0.012 | 1.41% |
> [0.7-0.8] | 0.017 | 2.28% | 0.013 | 1.71% |
>
> In addition to the aforementioned experiments, we also included an application scenario experiment on Question Answer Verification, which is also based on BERT. You can find the detailed introduction of the experiment in [22]. We compared the Top-K Token Retention with Attention Flow of FlowPrune and Attention Rollout, and organized the results in a manner similar to *Table 1* in Section 4.2.1. The results show that, compared to Attention Rollout, the FlowPrune method is much closer to the analysis results of Attention Flow, demonstrating FlowPrune outperforms Attention Rollout. The table below presents the results:
>
> |Max-n|FP Pretrained(Mean ± SE)|AR Pretrained(Mean ± SE)|FP Fine-tuned(Mean ± SE)|AR Fine-tuned(Mean ± SE)|
> |---|---|---|---|---|
> | n = 3 | 0.9805 ± 0.0111 | 0.6468 ± 0.0291 | 0.9610 ± 0.0154 | 0.8571 ± 0.0259 |
> | n = 5 | 0.9731 ± 0.0154 | 0.8448 ± 0.0214 | 0.9416 ± 0.0231 | 0.7631 ± 0.0221 |
> | n = 10 | 0.9701 ± 0.0172 | 0.6650 ± 0.0188 | 0.9342 ± 0.0260 | 0.7103 ± 0.0176 |
>
> **3. Difference between llava and llama.** For the differences in absolute error patterns between LLaMA and LLaVA in Fig. 4, we think this is cased by the distinct attention graphs of the two models. LLaMA is a unimodal language model, while LLaVA is a multimodal vision-language model, resulting in inherently different attention graph structures. We hypothesize that the higher errors observed in LLaVA stem from its large number of input vision tokens, which often contain redundant information and contribute to an abundance of low-attention values. As discussed in **4.Low Attention Regions.**, FlowPrune exhibits higher approximation error in low attention regions, thereby increasing the overall error in LLaVA. A similar phenomenon appears in our experiments with Qwen (unimodal) and Qwen-VL (multimodal), reinforcing the notion that attention graph modality plays a key role in shaping the error pattern.
>
> In contrast, both LLaMA and LLaVA show similar trends in *relative error*. As highlighted in Section 4.2, relative error is of greater practical importance than absolute error, particularly for interpretability and token-level ranking tasks based on attention flow.
>
> **4. Low Attention Regions.** As discussed in Limitations (lines 336–341), FlowPrune exhibits relatively larger approximation error in low-attention regions. Here, "low-attention regions" refer to token pairs with small flow values in the original attention flow, indicating weak attentional connections. In practice, such pairs are typically of limited interest. As noted in lines 247–251, researchers are primarily concerned with token pairs exhibiting large attention flow values, where FlowPrune achieves much lower error. Therefore, the observed approximation inaccuracies in low-attention areas have minimal impact on practical applications.
>
> **5. Theoretical Guarantees.** We apologize for any confusion caused by our use of the term *theoretical guarantees*. To clarify, we do not claim a general guarantee on the approximation error, as this depends heavily on the specific numerical structure of the model’s attention graph and is analytically intractable. Rather, our point is that FlowPrune is grounded in the Max-Flow Min-Cut Theorem and thus offers a principled approximation of attention flow as originally defined in [14]. In contrast, Attention Rollout relies on recursively multiplying attention matrices, which is heuristic in nature and diverges significantly from the original flow formulation. Therefore, the *guarantee* we refer to is that FlowPrune adheres to the theoretical definition of attention flow, rather than serving as a purely heuristic alternative.

---

> > ### Comment · Reviewer_suGC · 2025-08-08
> >
> > Thank you for providing the detailed response. The current version has addressed most of my concerns, and I will adjust my score as needed. To further improve the paper, I suggest including a brief discussion of well-known alternatives to the max-flow/min-cut approach, such as cut-preserving, spectral, or flow sparsifiers, and providing additional justification for why max-flow/min-cut is the most suitable choice in this context. It would also strengthen the paper to clarify your position on theoretical guarantees, as mentioned in your response, in the revised version.

---

> > > ### Author Response · Authors · 2025-08-09
> > >
> > > Thank you for your valuable feedback. We will carefully consider your suggestions and work to implement them into the revised version.

---

> ### Author Response · Authors · 2025-08-06
>
> Dear Reviewers:
>
> We hope this message finds you well.
>
> Thank you again for your valuable feedback on our submission. We have submitted our rebuttal and are eager to engage in a discussion before the phase concludes.
>
> We truly appreciate the time and effort you have dedicated to reviewing our paper. We believe the points you raised are crucial for improving our work and are eager to provide any further clarifications that may be needed.
>
> As the discussion phase is approaching its end, we would greatly appreciate the opportunity to provide any further clarifications and address any remaining questions you may have.
>
> Thank you for your time and dedication to the review process.
>
> Sincerely,
>
> Authors of Submission 18977

---

### Official Review · Reviewer_b9Ta · 2025-07-03

**Clarity:** 3
**Significance:** 3
**Originality:** 3
**Rating:** 4
**Confidence:** 4

**Summary:**

This paper focuses on the computational inefficiency of attention flow, an interpretability technique, when applied to large-scale transformer models. To handle this, the authors exploit two properties of transformers, i.e., the softmax activation in attention map and the strictly layered-structure. Based on these, they propose Edge Pruning to prune less informative token-to-token connections via a predefined threshold and Layer Compression to remove layers with minimal contribution to attention flow. Together, the two strategies reduce the size of attention graph, thereby mitigating the computational complexity. Experiments on LLaMA and LLaVA validate the efficiency of the proposed method.

**Questions:**

See weaknesses for details.

**Ethical Concerns:**

["NO or VERY MINOR ethics concerns only"]

**Limitations:**

None.

**Paper Formatting Concerns:**

None.

**Quality:**

3

**Strengths And Weaknesses:**

Strengths:
1.	The problem that FlowPrune investigates is of practical impact, which helps to reduce the running time for interpreting behaviors of large-scale transformer models.
2.	Multiple well-designed metrics are used to evaluate the efficiency and interpretation quality.
3.	The paper is well organized and easy to follow.


Weaknesses:
1.	The pruning strategy of FlowPrune’s is relatively static, especially the threshold used in the Edge Pruning. Besides, the method primarily relies on the properties of softmax sparsity and layer-wise structure, which may limit its effectiveness when applied to specialized Transformer variants (e.g., sparse or hybrid attention models).
2.	Lack of discussion of recent related work and the compared methods are not recent, e.g., the Attention Rollout in 2020.
3.	Is the proposed method sensitive to the edge pruning threshold? Besides, what is the memory complexity?
4.	Minor suggestions: “Max-Flow Min-Cut Theorem” and “Highest-Level Preflow Push algorithm” first appear without references. In addition, there are some syntax errors, e.g., “(Line 42) ..., [14] propose ...” and “(Line 94) ..., Push–Relabel methods [33], which maintain ...”

---

> ### Author Rebuttal · Authors · 2025-07-30
>
> Dear reviewer:
>
> Thank you for your constructive comments, Your valuable comments have been of considerable assistance to us. Then, to your concerns, we address the comments as follows:
>
> **1. Static Threshold.** We use a static threshold primarily because experimental results show that FlowPrune is not particularly sensitive to its value. Among the tested values, $1 \times 10^{-6}$ consistently performs well and offers a reasonable balance. Given this robustness, using a static threshold simplifies implementation without compromising performance.The table below illustrates the average absolute error for the LLaMA model under different edge pruning thresholds after retaining half of the layers (16 layers).
>
> |1e-8|1e-7|1e-6|1e-5|
> |---|---|---|---|
> |0.0137|0.0099|0.0097|0.0103|
>
> **2. Effectiveness.** Firstly, the dominant Large Language Models (LLMs) and Large Vision-Language Models (LVLMs) currently in use primarily employ standard Transformer blocks, which do not incorporate sparse or hybrid attention mechanisms. Secondly, and more importantly, most existing sparse or hybrid attention mechanisms,  such as those used in Longformer[A] and Natively Trainable Sparse  Attention (NSA)[B], also exhibit softmax sparsity and layer-wise  characteristics. The layer-wise characteristic is applicable to almost  all Transformer architectures. As for softmax sparsity, as long as softmax is utilized in the  architecture, softmax sparsity is inevitable. For example, Longformer  applies softmax to the sparsified attention matrix, and NSA selectively  applies softmax to different attention paths.
>
> [A]  Longformer: The long-document transformer.
>
> [B] Natively Trainable Sparse Attention for Efficient Transformer.
>
> **3. Related Studies.** While we may seem to overlook related studies about attention flow or attention rollout, this is because recent research has largely focused on *using* attention flow or rollout as tools for analyzing model behavior or supporting new algorithmic designs, rather than on *improving* the computation of attention flow itself. Thus, these methods are typically treated as fixed, with limited consideration given to its efficiency or scalability. However, FlowPrune is introduced precisely to fill this overlooked gap by offering a more efficient and principled method for attention flow analysis, facilitating its broader application in large-scale interpretability tasks.
>
> **4. Memory Complexity.**  The memory complexity of FlowPrune is proportional to the size of the attention graph, as its primary memory usage arises during graph construction. The attention graph contains $O(LN)$ nodes and $O(LN^2)$ edges, resulting in an overall memory complexity of $O(LN^2 + LN) = O(LN^2)$.
>
> **5. Minor suggestions.** We apologize for these grammar mistakes and missing citations and thank you for pointing them out. They will be corrected in revision to make overall writing meet the standards.

---

> ### Author Response · Authors · 2025-08-06
>
> Dear Reviewers:
>
> We hope this message finds you well.
>
> Thank you again for your valuable feedback on our submission. We have submitted our rebuttal and are eager to engage in a discussion before the phase concludes.
>
> We truly appreciate the time and effort you have dedicated to reviewing our paper. We believe the points you raised are crucial for improving our work and are eager to provide any further clarifications that may be needed.
>
> As the discussion phase is approaching its end, we would greatly appreciate the opportunity to provide any further clarifications and address any remaining questions you may have.
>
> Thank you for your time and dedication to the review process.
>
> Sincerely,
>
> Authors of Submission 18977

---

### Decision · Program_Chairs · 2025-09-17

**Decision:**

Accept (poster)

**Comment:**

This paper uses max-flow algorithms to analyse and improve the attention mechanism in transformers. The paper is found to be standing on solid grounds by the reviewers, and even the skeptic reviewer acknowledged that his concerns were responded well by the authors. I am wondering about the complexity of the algorithms and whether they are tangible for large networks. The authors should be more careful with their writing.